# Communication Efficient Federated Learning with Differentiated Aggregation

Peyman Gholami [1]   Hulya Seferoglu [1]

## Abstract

This paper focuses on reducing the communication cost of federated learning by exploring generalization bounds and representation learning. We first characterize a tighter generalization bound for one-round federated learning based on local clients' generalizations and heterogeneity of data distribution (non-iid scenario). We also characterize a generalization bound in R-round federated learning and its relation to the number of local updates (local stochastic gradient descents (SGDs)). Then, based on our generalization bound analysis and our representation learning interpretation of this analysis, we show for the first time that less frequent aggregations, hence more local updates, for the representation extractor (usually corresponds to initial layers) leads to the creation of more generalizable models, particularly for non-iid scenarios. We design a novel Federated Learning with Adaptive Local Steps (FedALS) algorithm based on our generalization bound and representation learning analysis. FedALS employs varying aggregation frequencies for different parts of the model, so reduces the communication cost. The paper is followed with experimental results showing the effectiveness of FedALS.

## 1. Introduction

Federated learning advocates that multiple clients collaboratively train machine learning models under the coordination of a parameter server (central aggregator) (McMahan et al., 2016). This approach has great potential for preserving the privacy of data stored at clients while simultaneously leveraging the computational capacities of all clients. Despite its promise, federated learning still suffers from high communication costs between clients and the parameter server.

In a federated learning setup, a parameter server oversees

[1]Department of Electrical and Computer Engineering, University of Illinois Chicago, Chicago, Illinois, United States. Correspondence to: Peyman Gholami <pghola2@uic.edu>.

Accepted to the Workshop on Advancing Neural Network Training at International Conference on Machine Learning (WANT@ICML 2024).

a global model and distributes it to participating clients. These clients then conduct local training using their own data. Then, the clients send their model updates to the parameter server, which aggregates them to a global model. This process continues until convergence. Exchanging machine learning models is costly, especially for large models, which are typical in today's machine learning applications (Konecný et al., 2016; Zhang et al., 2013; Barnes et al., 2020; Braverman et al., 2015). Furthermore, the uplink bandwidth of clients may be limited, time-varying and expensive. Thus, there is an increasing interest in reducing the communication cost of federated learning especially by taking advantage of multiple local updates also known as "Local SGD" (Stich, 2018; Stich & Karimireddy, 2019; Wang & Joshi, 2018). The crucial questions in this context are (i) how long clients shall do Local SGD, (ii) when they shall aggregate their local models, and (iii) which parts of the model shall be aggregated. The goal of this paper is to address these questions and reduce communication costs without hurting convergence.

The primary purpose of communication in federated learning is to periodically aggregate local models to reduce the consensus distance among clients. This practice helps maintain the overall optimization process on a trajectory toward global optimization. It is important to note that when the consensus distance among clients becomes substantial, the convergence rate reduces. This occurs as individual clients gradually veer towards their respective local optima without being synchronized with the models from other clients. This issue is amplified when the data distribution among clients is non-iid. It has been demonstrated that the consensus distance is correlated to (i) the randomness in each client's own dataset, which causes variation in consecutive local gradients, as well as (ii) the dissimilarity in loss functions among clients due to non-iidness (Stich & Karimireddy, 2019; Gholami & Seferoglu, 2024). More specifically, the consensus distance at iteration $t$ is defined as $\frac{1}{K}\sum_{k=1}^{K}\|\hat{\boldsymbol{\theta}}_t - \boldsymbol{\theta}_{k,t}\|^2$, where $\hat{\boldsymbol{\theta}}_t = \frac{1}{K}\sum_{k=1}^{K}\boldsymbol{\theta}_{k,t}$, $K$ is the number of clients, $\boldsymbol{\theta}_{k,t}$ is the local model at client $k$ at iteration $t$, and $\|\cdot\|^2$ is squared $l_2$ norm. Note that the consensus distance goes to zero when global aggregation is performed at each communication round. This makes the communication of models between clients and the parameter server crucial, but this introduces significant communication overhead. This paper

aims to reduce the communication overhead of federated learning through the following contributions.

*Contribution I: Improved Generalization Error Bound.* The generalization error of a learning model is defined as the difference between the model's empirical risk and population risks. (We provide a mathematical definition in Section 3). Existing approaches for training models mostly minimize the empirical risk or its variants. However, a small population risk is desired showing how well the model performs in the test phase as it denotes the loss that occurs when new samples are randomly drawn from the distribution. Note that a small empirical risk and a reduced generalization error correspond to a low population risk. Thus, there is an increasing interest in establishing an upper limit for the generalization error and understanding the underlying factors that affect the generalization error. The generalization error analysis is also important to quantitatively assess the generalization characteristics of trained models, provide reliable guarantees concerning their anticipated performance quality, and design new models and systems.

In this paper, we offer a tighter generalization bound compared to the state of the art (Barnes et al., 2022; Yagli et al., 2020; Sun et al., 2023) for one-round federated learning, considering local clients' generalizations and non-iidness (i.e., heterogeneous data distribution across the clients). Additionally, we characterize the generalization error bound in R-round federated learning.

*Contribution II: Representation Learning Interpretation.* Recent studies have demonstrated that the concept of representation learning is a promising approach to reducing the communication cost of federated learning (Collins et al., 2021). This is achieved by leveraging the shared representations in all clients' datasets. For example, let us consider a federated learning application for image classification, where different clients have datasets of different animals. Despite each client having a different dataset (one client has dog images, another has cat images, etc.), these images usually have common features such as an eye/ear shape. These shared features, typically extracted in the same way for different types of animals, require consistent layers of a neural network to extract them, whether the animal is a dog or a cat. As a result, these layers demonstrate similarity (i.e., less variation) across clients even when the datasets are non-iid. This implies that the consensus distance for this part of the model (feature extraction) is likely smaller. Based on these observations, our key idea is to reduce the aggregation frequency of the layers that show high similarity, where these layers are updated locally between consecutive aggregations. This approach would reduce the communication cost of federated learning as some layers are aggregated, hence their parameters are exchanged, less frequently. The next example scratches the surface of the problem for a toy

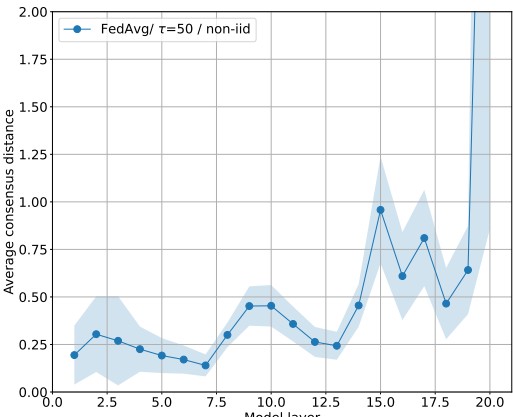

Figure 1: Average consensus distance over time for different layers, measured while training a ResNet-20 by FedAvg on CIFAR-10 with 5 clients with non-iid data distribution over clients (2 classes per client). The early layers responsible for extracting representations exhibit lower levels of consensus distance.

example.

**Example 1.** *We consider a federated learning setup of five clients with a central parameter server to train a ResNet-20 (He et al., 2015) on a heterogeneous partition of CIFAR-10 dataset (Krizhevsky, 2009). We use Federated Averaging (FedAvg) (McMahan et al., 2016) as an aggregation algorithm since it is the dominant algorithm in federated learning. We applied FedAvg with 50 local steps prior to each averaging step, denoted as $\tau = 50$. Non-iidness is introduced by allocating 2 classes to each client. Finally, we evaluate the quantity of the average consensus distance for each model layer during the optimization in Fig. 1. It is clear that the initial layers have smaller consensus distance as compared to the final layers. This is due to initial layers' role in extracting representations from input data and their higher similarity across clients.*

The above example indicates that initial layers show higher similarity, so they can be aggregated less frequently. Additionally, several empirical studies (Reddi et al., 2021; Yu et al., 2020) show that federated learning with multiple local updates per round learns a generalizable representation and is unexpectedly successful in non-iid settings. These studies encourage us to delve deeper into investigating how local updates and model aggregation frequency affect the model's representation extractor in terms of its generalization.

In this paper, based on our improved generalization bound analysis and our representation learning interpretation of this analysis, we showed for the first time that employing different frequencies of aggregation, *i.e.,* the number of local updates (local SGDs), for the representation extractor (typically corresponding to initial layers) and the head (final label prediction layers), leads to the creation of more

generalizable models particularly in non-iid scenarios.

*Contribution III: Design of FedALS.* We design a novel Federated Learning with Adaptive Local Steps (FedALS) algorithm based on our generalization error bound analysis and its representation learning interpretation. FedALS employs varying aggregation frequencies for different parts of the model.

*Contribution IV: Evaluation.* We evaluate the performance of FedALS using deep neural network model ResNet-20 for CIFAR-10, CIFAR-100 (Krizhevsky, 2009), SVHN (Netzer et al., 2011), and MNIST (Lecun et al., 1998) datasets. We also estimate the impact of FedALS on large language models (LLMs) in fine-tuning OPT-125M (Zhang et al., 2022) on the Multi-Genre Natural Language Inference (MultiNLI) corpus (Williams et al., 2018). We consider both iid and non-iid data distributions. Experimental results confirm that FedALS outperforms the baselines in terms of accuracy in non-iid setups while also saving on communication costs across all setups.

## 2. Related Work

There has been increasing interest in distributed learning recently, largely driven by Federated Learning. Several studies have highlighted that these algorithms achieve convergence to a global optimum or a stationary point of the overall objective, particularly in convex or non-convex scenarios (Stich & Karimireddy, 2019; Stich, 2018; Gholami & Seferoglu, 2024; Lian et al., 2018; Kairouz et al., 2019). However, it is widely accepted that communication cost is the major bottleneck for these techniques in large-scale optimization applications (Konecný et al., 2016; Lin et al., 2017). To tackle this issue, two primary strategies are put forth: the utilization of mini-batch parallel SGD, and the adoption of Local SGD. These approaches aim to enhance the equilibrium between computation and communication. Woodworth et al. (2020b;a) attempt to theoretically capture the distinction to comprehend under what circumstances Local SGD outperforms minibatch SGD.

Local SGD appears to be more intuitive compared to mini-batch SGD, as it ensures progress towards the optimum even in cases where workers are not communicating and employing a mini-batch size that is too large may lead to a decrease in performance (Lin et al., 2017). However, due to the fact that individual gradients for each worker are computed at distinct instances, this technique brings about residual errors. As a result, a compromise arises between reducing communication rounds and introducing supplementary errors into the gradient estimations. This becomes increasingly significant when data is unevenly distributed across nodes. There are several decentralized algorithms that have been shown to mitigate heterogeneity (Karimireddy et al., 2019; Liu et al., 2023) . One prominent example is the Stochastic Con-

trolled Averaging algorithm (SCAFFOLD) (Karimireddy et al., 2019), which addresses the node drift caused by non-iid characteristics of data distribution. They establish the notion that SCAFFOLD demonstrates a convergence rate at least equivalent to SGD, ensuring convergence even when dealing with highly non-iid datasets.

However, despite these factors, multiple investigations (Reddi et al., 2021; Yu et al., 2020; Lin et al., 2020; Gu et al., 2023), have noted that the model trained using FedAvg and incorporating multiple Local SGD per round exhibits unexpected effectiveness when subsequently fine-tuned for individual clients in non-iid FL setting. This implies that the utilization of FedAvg with several local updates proves effective in acquiring a valuable data representation, which can later be employed on each node for downstream tasks. Following this line of reasoning, our justification will be based on the argument that the Local SGD component of FedAvg contributes to improving performance in heterogeneous scenarios by facilitating the acquisition of models with enhanced generalizability.

An essential characteristic of machine learning systems is their capacity to extend their performance to novel and unseen data. This capacity, referred to as generalization, can be expressed within the framework of statistical learning theory. There has been a line of research to characterize generalization bound in FL (Wang & Ma, 2023; Mohri et al., 2019). More recently Barnes et al. (2022); Sun et al. (2023); Yagli et al. (2020) considered this problem and gave upper bounds on the expected generalization error for FL in iid setting in terms of the local generalizations of clients. This work demonstrates an improved dependence of $\frac{1}{K}$ on the number of nodes. Motivated by this work, we build our research foundation by analyzing generalization in a non-iid setting and use the derived insights to introduce FedALS, aiming to enhance conventional machine learning generalization.

Note that FedALS differs from exploiting shared representations for personalized federated learning, as discussed in Collins et al. (2021). In FedALS, we do not employ different models on different clients, as seen in personalized learning. Our proof demonstrates that increasing the number of local steps enhances generalization in the standard (single-model) federated learning setting.

## 3. Background and Problem Statement

### 3.1. Preliminaries and Notation

We consider that we have $K$ clients/nodes in our system, and each node has its own portion of the dataset. For example, node $k$ has a local dataset $S_k = \{z_{k,1}, ..., z_{k,n_k}\}$, where $z_{k,i} = (x_{k,i}, y_{k,i})$ is drawn from a distribution $\mathcal{D}_k$ over $\mathcal{X} \times \mathcal{Y}$ , where $\mathcal{X}$ is the input space and $\mathcal{Y}$ is the label space. We consider $\mathcal{X} \subseteq \mathbb{R}^d$ and $\mathcal{Y} \subseteq \mathbb{R}$ . The size of the

local dataset at node $k$ is $n_k$. The dataset across all nodes is defined as $S = \{S_1, ..., S_K\}$. Data distribution across the nodes could be independent and identically distributed (iid) or non-iid. In iid setting, we assume that $\mathcal{D}_1 = ... = \mathcal{D}_K = \mathcal{D}$ holds. On the other hand, non-iid setting covers all possible distributions and cases, where $\mathcal{D}_1 = ... = \mathcal{D}_K = \mathcal{D}$ does not hold.

We assume that $M_{\boldsymbol{\theta}} = \mathcal{A}(S)$ represents the output of a possibly stochastic function denoted as $\mathcal{A}(S)$, where $M_{\boldsymbol{\theta}} : \mathcal{X} \to \mathcal{Y}$ represents the learned model parameterized by $\boldsymbol{\theta}$. We consider a real-valued loss function denoted as $l(M_{\boldsymbol{\theta}}, \boldsymbol{z})$, which assesses the model $M_{\boldsymbol{\theta}}$ based on a sample $\boldsymbol{z}$.

### 3.2. Generalization Error

We first define an empirical risk on dataset $S$ as

$$R_{S}(M_{\boldsymbol{\theta}}) = \mathbb{E}_{k \sim \mathcal{K}} R_{S_k}(M_{\boldsymbol{\theta}}) = \mathbb{E}_{k \sim \mathcal{K}} \frac{1}{n_k} \sum_{i=1}^{n_k} l(M_{\boldsymbol{\theta}}, \boldsymbol{z}_{k,i}), \quad (1)$$

where $\mathcal{K}$ is an arbitrary distribution over nodes to weight different local risk contributions in the global risk. Specifically, $\mathcal{K}(k)$ represents the contribution of node $k$'s loss in the global loss. In the most conventional case, it is usually assumed to be uniform across all nodes, i.e., $\mathcal{K}(k) = \frac{1}{K}$ for all $k$. $R_{S_k}(M_{\boldsymbol{\theta}})$ is the empirical risk for model $M_{\boldsymbol{\theta}}$ on local dataset $S_k$. We further define a population risk for model $M_{\boldsymbol{\theta}}$ as

$$R(M_{\boldsymbol{\theta}}) = \mathbb{E}_{k \sim \mathcal{K}} R_k(M_{\boldsymbol{\theta}}) = \mathbb{E}_{k \sim \mathcal{K}, \boldsymbol{z} \sim \mathcal{D}_k} l(M_{\boldsymbol{\theta}}, \boldsymbol{z}), \quad (2)$$

where $R_k(M_{\boldsymbol{\theta}})$ is the population risk on node $k$'s data distribution.

Now, we can define the generalization error for dataset $S$ and function $\mathcal{A}(S)$ as

$$\Delta_{\mathcal{A}}(S) = R(\mathcal{A}(S)) - R_{S}(\mathcal{A}(S)). \quad (3)$$

The expected generalization error is expressed as $\mathbb{E}_{S} \Delta_{\mathcal{A}}(S)$, where $\mathbb{E}_{S}[\cdot] = \mathbb{E}_{\{S_k \sim \mathcal{D}_k^{n_k}\}_{k=1}^{K}}[\cdot]$ is used for the sake of notation convenience.

### 3.3. Federated Learning

We consider a federated learning scenario with $K$ nodes/clients and a centralized parameter server. The nodes update their localized models to minimize their empirical risk $R_{S_k}(M_{\boldsymbol{\theta}})$ on local dataset $S_k$, while the parameter server aggregates the local models to minimize the empirical risk $R_{S}(M_{\boldsymbol{\theta}})$. Due to connectivity and privacy constraints, the clients do not exchange their data with each other. One of the most widely used federated learning algorithms is FedAvg (McMahan et al., 2016), which we explain in detail next.

At round $r$ of FedAvg, each node $k$ trains its model $M_{\boldsymbol{\theta}_{k,r}} = \mathcal{A}_{k,r}(S_k)$ locally using the function/algorithm $\mathcal{A}_{k,r}$. The

local models $M_{\boldsymbol{\theta}_{k,r}}$ are transmitted to the central parameter server, which merges the received local models to aggregated model parameters $\hat{\boldsymbol{\theta}}_{r+1} = \hat{\mathcal{A}}(\boldsymbol{\theta}_{1,r}, ..., \boldsymbol{\theta}_{K,r})$, where $\hat{\mathcal{A}}$ is the aggregation function. In FedAvg, the aggregation function calculates an average, so the aggregated model is expressed as

$$\hat{\boldsymbol{\theta}}_{r+1} = \mathbb{E}_{k \sim \mathcal{K}} \, \boldsymbol{\theta}_{k,r}. \quad (4)$$

Subsequently, the aggregated model is transmitted to all nodes. This process continues for $R$ rounds. The final model after $R$ rounds of FedAvg is $\mathcal{A}(S)$.

The local models are usually trained using stochastic gradient descent (SGD) at each node. To reduce the communication cost needed between the nodes and the parameter server, each node executes multiple SGD steps using its local data after receiving an aggregated model from the parameter server. To be precise, we have the aggregated model parameters at round $r$ as $\hat{\boldsymbol{\theta}}_r$. Specifically, upon receiving $\hat{\boldsymbol{\theta}}_r$, node $k$ computes

$$\boldsymbol{\theta}_{k,r,t+1} = \boldsymbol{\theta}_{k,r,t} - \frac{\eta}{|\mathcal{B}_{k,r,t}|} \sum_{i \in \mathcal{B}_{k,r,t}} \nabla l(M_{\boldsymbol{\theta}_{k,r,t}}, \boldsymbol{z}_{k,i}) \quad (5)$$

for $t = 0, \ldots, \tau - 1$, where $\tau$ is the number of local SGD steps, $\boldsymbol{\theta}_{k,r,0}$ is defined as $\boldsymbol{\theta}_{k,r,0} = \hat{\boldsymbol{\theta}}_r$, $\eta$ is the learning rate, $\mathcal{B}_{k,r,t}$ is the batch of samples used in local step $t$ of round $r$ in node $k$, $\nabla$ is the gradient, and $|\cdot|$ shows the size of a set. Upon completing the local steps in round $r$, each node transmits $\boldsymbol{\theta}_{k,r} = \boldsymbol{\theta}_{k,r,\tau}$ to the parameter server to calculate $\hat{\boldsymbol{\theta}}_{r+1}$ as in (4).

### 3.4. Representation Learning

Our approach for analyzing the generalization error bounds for federated learning, by specifically focusing on FedAvg, uses representation learning, which we explain next.

We consider a class of models that consist of a representation extractor (*e.g.,* ResNet). Let $\boldsymbol{\theta}$ be the model $M_{\boldsymbol{\theta}}$'s parameters. We can decompose $\boldsymbol{\theta}$ into two sets: $\boldsymbol{\phi}$ containing the representation extractor's parameters and $\boldsymbol{h}$ containing the head parameters, *i.e.,* $\boldsymbol{\theta} = [\boldsymbol{\phi}, \boldsymbol{h}]$. $M_{\boldsymbol{\phi}}$ is a function that maps from the original input space to some feature space, *i.e.,* $M_{\boldsymbol{\phi}} : \mathbb{R}^d \to \mathbb{R}^{d'}$, where usually $d' \ll d$. The function $M_{\boldsymbol{h}}$ performs a low complexity mapping from the representation space to the label space, which can be expressed as $M_{\boldsymbol{h}} : \mathbb{R}^{d'} \to \mathbb{R}$.

For any $\boldsymbol{x} \in \mathcal{X}$, the output of the model is $M_{\boldsymbol{\theta}}(\boldsymbol{x}) = (M_{\boldsymbol{\phi}} \circ M_{\boldsymbol{h}})(\boldsymbol{x}) = M_{\boldsymbol{h}}(M_{\boldsymbol{\phi}}(\boldsymbol{x}))$. For instance, if $M_{\boldsymbol{\theta}}$ is a neural network, $M_{\boldsymbol{\phi}}$ represents several initial layers of the network, which are typically designed to extract meaningful representations from the neural network's input. On the other hand, $M_{\boldsymbol{h}}$ denotes the final few layers that lead to the network's output.

# 4. Improved Generalization Bounds

In this section, we derive generalization bounds for FedAvg based on clients' local generalization performances in a general non-iid setting for the first time in the literature. First, we start with one-round FedAvg and analyze its generalization bound. Then, we extend our analysis to $R-$round FedAvg.

## 4.1. One-Round Generalization Bound

In the following theorem, we determine the generalization bound for one round of FedAvg.

**Theorem 4.1.** *Let $l(M_{\boldsymbol{\theta}}, \boldsymbol{z})$ be $\mu$-strongly convex and $L$-smooth in $M_{\boldsymbol{\theta}}$. $M_{\boldsymbol{\theta}_k} = \mathcal{A}_k(\boldsymbol{S}_k)$ represents the model obtained from Empirical Risk Minimization (ERM) algorithm on local dataset $\boldsymbol{S}_k$, i.e., $M_{\boldsymbol{\theta}_k} = \arg\min_M \sum_{i=1}^{n_k} l(M, \boldsymbol{z}_{k,i})$, and $M_{\hat{\boldsymbol{\theta}}} = \mathcal{A}(\boldsymbol{S})$ is the model after one round of FedAvg ($\hat{\boldsymbol{\theta}} = \mathbb{E}_{k \sim \mathcal{K}} \boldsymbol{\theta}_k$). Then, the expected generalization error is*

$$\mathbb{E}_{\boldsymbol{S}} \Delta_{\mathcal{A}}(\boldsymbol{S}) \le \mathbb{E}_{k \sim \mathcal{K}} \left[ \frac{L\mathcal{K}(k)^2}{\mu} \underbrace{\mathbb{E}_{\boldsymbol{S}_k} \Delta_{\mathcal{A}_k}(\boldsymbol{S}_k)}_{\textit{Expected local generalization}} \right. \tag{6}$$

$$\left. + 2\sqrt{\frac{L}{\mu}} \mathcal{K}(k) \left( \underbrace{\mathbb{E}_{\boldsymbol{S}} \, \delta_{k,\mathcal{A}}(\boldsymbol{S})}_{\textit{Expected non-iidness}} \underbrace{\mathbb{E}_{\boldsymbol{S}_k} \Delta_{\mathcal{A}_k}(\boldsymbol{S}_k)}_{\textit{Expected local generalization}} \right)^{\frac{1}{2}} \right],$$

*where $\delta_{k,\mathcal{A}}(\boldsymbol{S}) = R_{\boldsymbol{S}_k}(\mathcal{A}(\boldsymbol{S})) - R_{\boldsymbol{S}_k}(\mathcal{A}_k(\boldsymbol{S}_k))$ indicates the level of non-iidness at client $k$ for function $\mathcal{A}$ on dataset $\boldsymbol{S}$.*

*Proof:* The proof of Theorem 4.1 is provided in Appendix A of the supplementary materials. □

*Remark 4.2.* We note that this theorem and its proof assume that all clients participate in learning. The other scenario is that not all clients participate in the learning procedure. We can consider the following two cases when not all clients participate in the learning procedure.

*Case I:* Sampling $\hat{K}$ clients with replacement based on distribution $\mathcal{K}$, followed by averaging the local models with equal weights.

*Case II:* Sampling $\hat{K}$ clients without replacement uniformly at random, then performing weighted averaging of local models. Here, the weight of client $k$ is rescaled to $\frac{\mathcal{K}(k)K}{\hat{K}}$.

The generalization error results in these cases are affected by substituting $\frac{1}{\hat{K}}$ and $\frac{\mathcal{K}(k)K}{\hat{K}}$ instead of $\mathcal{K}(k)$ in (6) for cases I and II, respectively. The detailed proof is provided in Appendix C.

**Discussion.** Note that there are two terms in the generalization error bound: (i) local generalization of each client that shows more generalizable local models lead to a better generalization of the aggregated model, (ii) non-iidness of each client which deteriorates generalization. Theorem 4.1

reveals a factor of $\mathcal{K}(k)^2$ for the first term, which is the sole term in the iid setting. For example, in the uniform case ($\mathcal{K}(k) = \frac{1}{K}$), we will observe an improvement with a factor of $\frac{1}{K^2}$ for the iid case. This represents an enhancement compared to the state of the art (Barnes et al., 2022; Sun et al., 2023; Yagli et al., 2020), which only demonstrates a factor of $\frac{1}{K}$. As a result, after the averaging process carried out by the central parameter server, the generalization error is reduced by a factor of $\mathcal{K}(k)^2$ in iid case.

On the other hand, we do not see a similar behavior in non-iid case. In other words, the expected generalization error bound does not necessarily decrease with averaging. These results show why FedAvg works well in iid setup, but not necessarily in non-iid setup. This observation motivates us to design a new federated learning approach for non-iid setup. The question in this context is what should be the new federated learning design. To answer this question, we analyze $R-$round generalization bound in the next section.

## 4.2. $R-$Round Generalization Bound

In this setup, after $R$ rounds, there is a sequence of weights $\{\hat{\boldsymbol{\theta}}_r\}_{r=1}^R$ and the final model is $\hat{\boldsymbol{\theta}}_R$. We consider that at round $r$, each node constructs its updated model as in (5) by taking $\tau$ gradient steps starting from $\hat{\boldsymbol{\theta}}_r$ with respect to $\tau$ random mini-batches $Z_{k,r} = \bigcup\{\mathcal{B}_{k,r,t}\}_{t=0}^{\tau-1}$ drawn from the local dataset $\boldsymbol{S}_k$. For this type of iterative algorithm, we consider the following averaged empirical risk

$$\frac{1}{R} \sum_{r=1}^R \mathbb{E}_{k \sim \mathcal{K}} \left[ \frac{1}{|Z_{k,r}|} \sum_{i \in Z_{k,r}} l(M_{\hat{\boldsymbol{\theta}}_r}, \boldsymbol{z}_{k,i}) \right]. \tag{7}$$

The corresponding generalization error, $\Delta_{FedAvg}(\boldsymbol{S})$, is

$$\frac{1}{R} \sum_{r=1}^R \mathbb{E}_{k \sim \mathcal{K}} \left[ \mathbb{E}_{\boldsymbol{z} \sim \mathcal{D}_k} l(M_{\hat{\boldsymbol{\theta}}_r}, \boldsymbol{z}) - \frac{1}{|Z_{k,r}|} \sum_{i \in Z_{k,r}} l(M_{\hat{\boldsymbol{\theta}}_r}, \boldsymbol{z}_{k,i}) \right]. \tag{8}$$

Note that the expression in (8) differs slightly from the end-to-end generalization error that would be obtained by considering the final model $M_{\hat{\boldsymbol{\theta}}_R}$ and the entire dataset $\boldsymbol{S}$. More specifically, (8) is an average of the generalization errors measured at each round, similar to (Barnes et al., 2022)). We anticipate that the generalization error diminishes with the increasing number of data samples, so this generalization error definition yields to a more cautious upper limit and serves as a sensible measure. The next theorem characterizes the expected generalization error bounds for $R-$Round FedAvg in iid and non-iid settings.

**Theorem 4.3.** *Let $l(M_{\boldsymbol{\theta}}, \boldsymbol{z})$ be $\mu$-strongly convex and $L$-smooth in $M_{\boldsymbol{\theta}}$. Local models at round $r$ are calculated by doing $\tau$ local gradient descent steps (5), and the local gradient variance is bounded by $\sigma^2$, i.e., $\mathbb{E}_{\boldsymbol{z} \sim \mathcal{D}_k} \|\nabla l(M_{\boldsymbol{\theta}}, \boldsymbol{z}) - \mathbb{E}_{\boldsymbol{z} \sim \mathcal{D}_k} \nabla l(M_{\boldsymbol{\theta}}, \boldsymbol{z})\|^2 \le \sigma^2$. The aggregated model at*

*round $r$, $M_{\hat{\theta}_r}$, is obtained by performing FedAvg, and the data points used in round $r$ (i.e., $Z_{k,r}$) are sampled without replacement. Then the average generalization error, $\mathbb{E}_{\boldsymbol{S}} \, \Delta_{FedAvg}(\boldsymbol{S})$, is upper bounded by*

$$\frac{1}{R} \sum_{r=1}^{R} \mathbb{E}_{k \sim \mathcal{K}} \left[ \frac{2L\mathcal{K}(k)^2}{\mu} A + \sqrt{\frac{8L}{\mu}} \mathcal{K}(k)(AB)^{\frac{1}{2}} \right], \quad (9)$$

*where $A = \tilde{O}\left( \sqrt{\frac{\mathcal{C}(M_{\boldsymbol{\theta}})}{|Z_{k,r}|}} + \frac{\sigma^2}{\mu\tau} + \frac{L}{\mu} \right)$, $B = \tilde{O}\left( \mathbb{E}_{\{Z_{k,r}\}_{k=1}^K} \delta_{k,\mathcal{A}}(\{Z_{k,r}\}_{k=1}^K) + \frac{\sigma^2}{\mu\tau} + \frac{L}{\mu} \right)$, $\tilde{O}$ hides constants and poly-logarithmic factors, and $\mathcal{C}(M_{\boldsymbol{\theta}})$ shows the complexity of the model class of $M_{\boldsymbol{\theta}}$.*

*Proof:* The proof of Theorem 4.3 is provided in Appendix B of the supplementary materials. □

The generalization error bound in (9) depends on the following parameters: (i) number of rounds; $R$, (ii) number of samples used in every round; $|Z_{k,r}|$, (iii) the complexity of the model class; $\mathcal{C}(M_{\boldsymbol{\theta}})$, non-iidness, $\delta_{k,\mathcal{A}}(\{Z_{k,r}\}_{k=1}^K)$, number of local steps in each round; $\tau$. We note that (9) also depends on $K$ (more specifically $\mathcal{K}$), but this dependence is similar to the discussion we had for one-round generalization, so we skip it here.

## 5. Representation Learning Interpretation of R-Round Generalization Bound

The complexity of the model class and the number of samples and local steps used in every round are crucial to minimizing the generalization error bound especially in non-iid case in (9) where the generalization error bound is loose in comparison to iid setup.

Some common complexity measures in the literature include the number of parameters (classical VC Dimension (Shalev-Shwartz & Ben-David, 2014)), parameter norms (*e.g.,* $l_1, l_2$, spectral) (Bartlett, 1997), or other potential complexity measures (Lipschitzness, Sharpness, . . . ) (Neyshabur et al., 2017; Dziugaite & Roy, 2017; Nagarajan & Kolter, 2019; Wei & Ma, 2019; Norton & Royset, 2019; Foret et al., 2021). Independent from a specific complexity measure, a model in representation learning can be divided into two parts: (i) $M_{\boldsymbol{\phi}}$, which is the representation extractor, and (ii) $M_{\boldsymbol{h}}$, a simple head which maps the representation to an output. The complexities of these parts follow $\mathcal{C}(M_{\boldsymbol{h}}) \ll \mathcal{C}(M_{\boldsymbol{\phi}})$.

Our key intuition in this paper is that we can *reduce the aggregation frequency of $M_{\boldsymbol{\phi}}$, which leads to a larger $\tau$ and $|Z_{k,r}|$, hence smaller generalization error bound according to (9).*[1]

As seen, there is a nice trade-off between aggregation fre-

---

[1] We do not reduce the aggregation frequency of $M_{\boldsymbol{h}}$ as its complexity, so its contribution to generalization error, is small.

---

**Algorithm 1** FedALS

**Input**: Initial model $\{\boldsymbol{\theta}_{k,0,0} = [\boldsymbol{\phi}_{k,0,0}, \boldsymbol{h}_{k,0,0}]\}_{k=1}^{K}$, learning rate $\eta$, number of local steps for the head $\tau$, adaptation coefficient $\alpha$.

1: **for** Round $r$ in $0, ..., R-1$ **do**
2:     **for** Node $k$ in $1, ..., K$ **in parallel do**
3:         **for** Local step $t$ in $0, ..., \tau-1$ **do**
4:             Sample the batch $\mathcal{B}_{k,r,t}$ from $\mathcal{D}_k$.
5:             $\boldsymbol{\theta}_{k,r,t+1} = \boldsymbol{\theta}_{k,r,t} - \frac{\eta}{|\mathcal{B}_{k,r,t}|} \sum_{i \in \mathcal{B}_{k,r,t}} \nabla l(M_{\boldsymbol{\theta}_{k,r,t}}, \boldsymbol{z}_{k,i})$
6:             **if** $\mod (r\tau + t, \tau) = 0$ **then**
7:                 $\boldsymbol{h}_{k,r,t} \leftarrow \frac{1}{K} \sum_{k=1}^{K} \boldsymbol{h}_{k,r,t}$
8:             **else if** $\mod (r\tau + t, \alpha\tau) = 0$ **then**
9:                 $\boldsymbol{\phi}_{k,r,t} \leftarrow \frac{1}{K} \sum_{k=1}^{K} \boldsymbol{\phi}_{k,r,t}$
10:         $\boldsymbol{\theta}_{k,r+1,0} = \boldsymbol{\theta}_{k,r,\tau}$
11: **return** $\hat{\boldsymbol{\theta}}_R = \frac{1}{K} \sum_{k=1}^{K} \boldsymbol{\theta}_{k,R,0}$

---

quency of $M_{\boldsymbol{\phi}}$ and population risk. In the next section, we design our Federated Learning with Adaptive Local Steps (FedALS) algorithm by taking into account this trade-off.

*Remark* 5.1. We note that the aggregation frequency of $M_{\boldsymbol{\phi}}$ cannot be reduced arbitrarily, as it would increase the empirical risk. It is proven that the convergence rate of the optimization problem of ERM in a general non-iid setting for a non-convex loss function is $O\left( \frac{\tau}{T} + \left(\frac{\tau}{T}\right)^{\frac{2}{3}} + \frac{1}{\sqrt{T}} \right)$ (Koloskova et al., 2020). Here $T$ is the total number of iterations, *i.e.,* $T = \tau R$.

## 6. FedALS: Federated Learning with Adaptive Local Steps

Theorem 4.3 and our key intuition above demonstrate that more local SGD steps (less aggregations at the parameter server) are necessary for representation extractor $M_{\boldsymbol{\phi}}$ as compared to the model's head $M_{\boldsymbol{h}}$ to reduce generalization error bound. This approach, since it will reduce the aggregation frequency of $M_{\boldsymbol{\phi}}$, will also reduce the communication cost of federated learning.

The main idea of FedALS is to maintain a uniform generalization error across both components ($M_{\boldsymbol{\phi}}$ and $M_{\boldsymbol{h}}$) of the model. This can be achieved if $\tau_{M_{\boldsymbol{\phi}}}$ is set larger than $\tau_{M_{\boldsymbol{\phi}}}$, where $\tau_M$ denotes the number of local iterations in a single round for the model $M$ while $\tau_{M_{\boldsymbol{\phi}}}$ and $\tau_{M_{\boldsymbol{h}}}$ are the corresponding number of local iterations for $M_{\boldsymbol{\phi}}$ and $M_{\boldsymbol{h}}$, respectively. Following this approach, we designed FedALS in Algorithm 1.

FedALS in Algorithm 1 divides the model into two parts: (i) the representation extractor, denoted as $M_{\boldsymbol{\phi}}$, and (ii) the head, denoted as $M_{\boldsymbol{h}}$. Additionally, we introduce the parameter $\alpha = \frac{\tau_{M_{\boldsymbol{\phi}}}}{\tau_{M_{\boldsymbol{h}}}}$ as an adaptation coefficient, which can be regarded as a hyperparameter for estimating the true

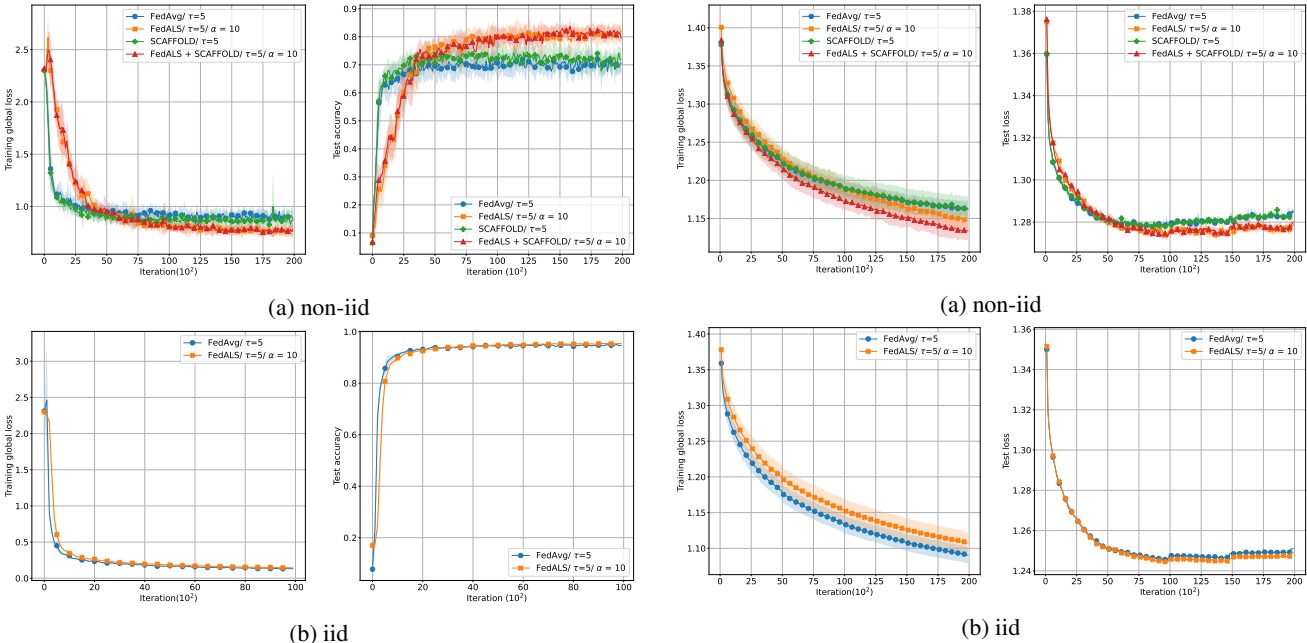

Figure 2: Training ResNet-20 on SVHN.

Figure 3: Fine-tuning OPT-125M on MultiNLI.

ratio. Note that this ratio depends on $\mathcal{C}(M_\phi)$ and $\mathcal{C}(M_h)$, and determining these values is not straightforward.

## 7. Experimental Results

In this section, we assess the performance of FedALS using ResNet-20 as a deep neural network architecture and OPT-125M as a large language model. We treat the convolutional layers of ResNet-20 as the representation extractor and the final dense layers as the model head. For OPT-125M, we consider the first 10 layers of the model as the representation extractor. We used the datasets CIFAR-10, CIFAR-100, SVHN, and MNIST for image classification and the Multi-Genre Natural Language Inference (MultiNLI) corpus for the LLM. The experimentation was conducted on a network consisting of five nodes alongside a central server. For image classification, we utilized a batch size of $64$ per node. SGD with momentum was employed as the optimizer, with the momentum set to $0.9$, and the weight decay to $10^{-4}$. For the LLM fine-tuning, we employed a batch size of 16 sentences from the corpus, and the optimizer used was AdamW. In all the experiments, to perform a grid search for the learning rate, we conducted each experiment by multiplying and dividing the learning rate by powers of two, stopping each experiment after reaching a local optimum learning rate. We repeat each experiment 20 times and present the error bars associated with the randomness of the optimization. In every figure, we include the average and standard deviation error bars.

### 7.1. FedALS in non-iid Setting

In this section, we allocate the dataset to nodes using a non-iid approach. For image classification, we initially sorted

the data based on their labels and subsequently divided it among nodes following this sorted sequence. In MultiNLI, we sorted the sentences based on their genre.

In this scenario, we can observe in Fig. 2a, 3a the anticipated performance improvement through the incorporation of different local steps across the model. By utilizing parameters $\tau = 5$ and $\alpha = 10$ in FedALS, it becomes apparent that aggregation and communication costs are reduced as compared to FedAvg with the same $\tau$ value of 5. This implies that the initial layers perform aggregation at every 50 iterations. This reduction in the number of communications is accompanied by enhanced model generalization stemming from the larger number of local steps in the initial layers, which contributes to an overall performance enhancement. Thus, our approach in FedALS is beneficial for both communication efficiency and enhancing model generalization performance simultaneously.

### 7.2. FedALS in iid Setting

The results for the iid setting are presented in Fig. 2b, 3b. In order to obtain these results, the data is shuffled, and then evenly divided among nodes. We note that in this situation, the performance improvement of FedALS is negligible. This is expected since there is a factor of $\frac{1}{K^2}$ in the generalization in this case, ensuring that we will have nearly the same population risk as the empirical risk. Therefore, the deciding factor here is the optimization of the empirical risk, which is improved with a smaller $\tau$ as discussed in Remark 5.1. Thus, the improvement of the generalization error using the FedALS approach is negligible in this setup.

Table 1: Test Performance for 5 nodes FL; Accuracy after training ResNet-20 and test loss after fine-tuning OPT-125M in iid and non-iid settings with $\tau = 5$ and $\alpha = 10$.

| MODEL/DATASET | FEDAVG | | FEDALS | | SCAFFOLD | FEDALS + SCAFFOLD |
|---|---|---|---|---|---|---|
| | IID | NON-IID | IID | NON-IID | NON-IID | NON-IID |
| RESNET-20/SVHN | $0.9476 \pm 0.0016$ | $\mathbf{0.7010 \pm 0.0330}$ | $0.9541 \pm 0.0021$ | $\mathbf{0.8117 \pm 0.0214}$ | $0.7180 \pm 0.04016$ | $0.8097 \pm 0.0271$ |
| RESNET-20/CIFAR-10 | $0.8761 \pm 0.0091$ | $\mathbf{0.4651 \pm 0.0071}$ | $0.8865 \pm 0.0021$ | $\mathbf{0.5224 \pm 0.0365}$ | $0.4538 \pm 0.0706$ | $0.5121 \pm 0.0098$ |
| RESNET-20/CIFAR-100 | $0.5999 \pm 0.0068$ | $\mathbf{0.4177 \pm 0.0143}$ | $0.6120 \pm 0.0052$ | $\mathbf{0.4863 \pm 0.0224}$ | $0.4124 \pm 0.0183$ | $0.4820 \pm 0.0129$ |
| RESNET-20/MNIST | $0.9906 \pm 0.0001$ | $\mathbf{0.7967 \pm 0.0635}$ | $0.9914 \pm 0.0001$ | $\mathbf{0.8208 \pm 0.0363}$ | $0.8121 \pm 0.0546$ | $0.7827 \pm 0.1184$ |
| OPT-125M/MULTINLI | $1.2503 \pm 0.0003$ | $\mathbf{1.2842 \pm 0.0016}$ | $1.2478 \pm 0.0005$ | $\mathbf{1.2773 \pm 0.0019}$ | $1.2839 \pm 0.0009$ | $1.2782 \pm 0.0018$ |

## 7.3. Compared to and Complementing SCAFFOLD

Karimireddy et al. (2019) introduced an innovative technique called SCAFFOLD, which employs some control variables for variance reduction to address the issue of "client-drift" in local updates. This drift happens when data is heterogeneous (non-iid), causing individual nodes/clients to converge towards their local optima rather than the global optima. While this approach is a significant theoretical advancement in achieving independence from loss function disparities among nodes, it hinges on the assumption of smoothness in the loss functions, which might not hold true for practical deep learning problems in the real world. Additionally, since SCAFFOLD requires the transmission of control variables to the central server, which is of the same size as the models themselves, it results in approximately twice the communication overhead when compared to FedAvg.

Let us consider Fig. 2a, 3a to notice that in real-world deep learning situations, FedALS enhances performance significantly, while SCAFFOLD exhibits slight improvements in specific scenarios. Moreover, we integrated FedALS and SCAFFOLD to concurrently leverage both approaches. The results of the test accuracy in different datasets are summarized in Table 1.

## 7.4. The Role of $\alpha$ and Communication Overhead

As shown in Table 2, it becomes evident that when we increase $\alpha$ from 1 (FedAvg), we initially witness an enhancement in accuracy owing to improved generalization. However, beyond a certain threshold ($\alpha = 10$), further increment in $\alpha$ ceases to contribute to performance improvement. This is due to the adverse impact of a high number of local steps on optimization performance indicated in Remark 5.1. The trade-off we discussed in the earlier sections is evident in this context. We have also demonstrated the impact of FedALS on the communication overhead in this table.

## 7.5. Different Combinations of $\phi, h$

In Table 3, we have presented the results of our experiments, illustrating how different combinations of $\phi$ and $h$ influence the model performance in FedALS. The parameter $L$ indicates the number of layers in the model considered as the representation extractor ($\phi$), while the remaining layers are considered as $h$. We observe that for ResNet-20, choosing $\phi$ to be the first 16 layers and performing less aggregation

Table 2: The accuracy and communication overhead per client after training ResNet-20 in non-iid setting with $\tau = 5$ and variable $\alpha$.

| VALUE OF $\alpha$ | DATASET | | # OF COMMUNICATED |
|---|---|---|---|
| | SVHN | CIFAR-10 | PARAMETERS |
| 1 | $0.7010 \pm 0.0330$ | $0.4651 \pm 0.0071$ | $2.344B$ |
| 5 | $0.8107 \pm 0.0278$ | $0.5201 \pm 0.0302$ | $0.473B$ |
| 10 | $\mathbf{0.8117 \pm 0.0214}$ | $\mathbf{0.5224 \pm 0.0365}$ | $0.239B$ |
| 25 | $0.7201 \pm 0.1565$ | $0.3814 \pm 0.0641$ | $0.099B$ |
| 50 | $0.6377 \pm 0.0520$ | $0.2853 \pm 0.0641$ | $0.052B$ |
| 100 | $0.5837 \pm 0.0715$ | $0.2817 \pm 0.032$ | $0.029B$ |

Table 3: Different Combinations of $\phi, h$ for training ResNet-20 in non-iid setting with $\tau = 5$, $\alpha = 10$.

| VALUE OF $L$ | DATASET | | |
|---|---|---|---|
| | SVHN | CIFAR-10 | CIFAR-100 |
| 20 | $0.6991 \pm 0.0160$ | $0.4383 \pm 0.0423$ | $0.4781 \pm 0.0123$ |
| 16 | $\mathbf{0.7112 \pm 0.0471}$ | $\mathbf{0.4687 \pm 0.0111}$ | $\mathbf{0.4782 \pm 0.0087}$ |
| 12 | $0.6760 \pm 0.0474$ | $0.4125 \pm 0.0283$ | $0.4249 \pm 0.0143$ |
| 8 | $0.6381 \pm 0.0428$ | $0.3779 \pm 0.03451$ | $0.4085 \pm 0.0094$ |
| 4 | $0.6339 \pm 0.0446$ | $0.3730 \pm 0.0310$ | $0.4183 \pm 0.0108$ |
| 1 | $0.6058 \pm 0.0197$ | $0.4013 \pm 0.0308$ | $0.3880 \pm 0.0305$ |

for them seems to be the most effective option.

## 8. Acknowledgments

This work was supported in part by ARL under Grant W911NF-2120272, and in part by NSF under Grant CCF-1942878, Grant CNS-2148182, and Grant CNS-2112471.

## 9. Conclusion

In this paper, we first characterized generalization error bound for one- and R-round federated learning. One-round generalization bound is tighter than the state of the art. Based on our improved generalization bound analysis and our representation learning interpretation of this analysis, we showed for the first time that less frequent aggregations, hence more local updates, for the representation extractor (usually corresponds to initial layers) leads to the creation of more generalizable models, particularly for non-iid scenarios. This insight led us to develop the FedALS algorithm, which centers around the concept of increasing local steps for the initial layers of the deep learning model while conducting more averaging for the final layers. The experimental results demonstrated the effectiveness of FedALS in heterogeneous setups.

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

## A. Proof of Theorem 4.1

We first state and prove the following lemma that will be used in the proof of Theorem 4.1.

**Lemma A.1** (Leave-one-out). *[Expansion of Theorem 1 in (Barnes et al., 2022)]*

*Let $S_k' = (z_{k,1}', ..., z_{k,n_k}')$, where $z_{k,i}'$ is sampled from $\mathcal{D}_k$. Denote $S^{(k)} = (S_1, ..., S_k', ..., S_K)$. Then*

$$\mathbb{E}_{\{S_k \sim \mathcal{D}_k^{n_k}\}_{k=1}^K} \Delta_{\mathcal{A}}(S) = \mathbb{E}_{k \sim \mathcal{K}, \{S_k, S_k' \sim \mathcal{D}_k^{n_k}\}_{k=1}^K} \left[ \frac{1}{n_k} \sum_{i=1}^{n_k} \left( l(\mathcal{A}(S), z_{k,i}') - l(\mathcal{A}(S^{(k)}), z_{k,i}') \right) \right]. \tag{10}$$

*Proof.* We have

$$\mathbb{E}_{\{S_k \sim \mathcal{D}_k^{n_k}\}_{k=1}^K} R(\mathcal{A}(S)) = \mathbb{E}_{k \sim \mathcal{K}, \{S_k, S_k' \sim \mathcal{D}_k^{n_k}\}_{k=1}^K} l(\mathcal{A}(S), z_{k,i}'). \tag{11}$$

Also, observe that

$$\mathbb{E}_{\{S_k \sim \mathcal{D}_k^{n_k}\}_{k=1}^K} R_S(\mathcal{A}(S)) = \mathbb{E}_{k \sim \mathcal{K}, \{S_k \sim \mathcal{D}_k^{n_k}\}_{k=1}^K} \left[ \frac{1}{n_k} \sum_{i=1}^{n_k} l(\mathcal{A}(S), z_{k,i}) \right] \tag{12}$$

$$= \mathbb{E}_{k \sim \mathcal{K}, \{S_k, S_k' \sim \mathcal{D}_k^{n_k}\}_{k=1}^K} \left[ \frac{1}{n_k} \sum_{i=1}^{n_k} l(\mathcal{A}(S^{(k)}), z_{k,i}') \right]. \tag{13}$$

Putting 11, and 13 together, and by the definition of the expected generalization error, we get

$$\mathbb{E}_{\{S_k \sim \mathcal{D}_k^{n_k}\}_{k=1}^K} \Delta_{\mathcal{A}}(S) = \mathbb{E}_{k \sim \mathcal{K}, \{S_k, S_k' \sim \mathcal{D}_k^{n_k}\}_{k=1}^K} \left[ \frac{1}{n_k} \sum_{i=1}^{n_k} \left( l(\mathcal{A}(S), z_{k,i}') - l(\mathcal{A}(S^{(k)}), z_{k,i}') \right) \right]. \tag{14}$$

$\square$

In the following lemma, we establish a fundamental generalization bound for a single round of ERM and FedAvg. (Theorem 4.1).

**Theorem A.2.** *Let $l(M_{\boldsymbol{\theta}}, z)$ be $\mu$-strongly convex and $L$-smooth in $M_{\boldsymbol{\theta}}$, $M_{\boldsymbol{\theta}_k} = \mathcal{A}_k(S_k)$ represents the model obtained from Empirical Risk Minimization (ERM) algorithm on local dataset $S_k$, i.e., $M_{\boldsymbol{\theta}_k} = \arg\min_M \sum_{i=1}^{n_k} l(M, z_{k,i})$, and $M_{\hat{\boldsymbol{\theta}}} = \mathcal{A}(S)$ is the model after one round of FedAvg ($\hat{\boldsymbol{\theta}} = \mathbb{E}_{k \sim \mathcal{K}} \boldsymbol{\theta}_k$). Then, the expected generalization error, $\mathbb{E}_{\{S_k \sim \mathcal{D}_k^{n_k}\}_{k=1}^K} \Delta_{\mathcal{A}}(S)$, is bounded by*

$$\mathbb{E}_{k \sim \mathcal{K}} \left[ \frac{L\mathcal{K}(k)^2}{\mu} \underbrace{\mathbb{E}_{\{S_k \sim \mathcal{D}_k^{n_k}\}} \Delta_{\mathcal{A}_k}(S_k)}_{\textit{Expected local generalization}} + 2\sqrt{\frac{L}{\mu}} \mathcal{K}(k) \underbrace{\sqrt{\mathbb{E}_{\{S_k \sim \mathcal{D}_k^{n_k}\}_{k=1}^K} \delta_{k,\mathcal{A}}(S)}}_{\textit{Root of expected non-iidness}} \underbrace{\sqrt{\mathbb{E}_{\{S_k \sim \mathcal{D}_k^{n_k}\}} \Delta_{\mathcal{A}_k}(S_k)}}_{\textit{Root of expected local generalization}} \right], \tag{15}$$

*where $\delta_{k,\mathcal{A}}(S) = \left[ R_{S_k}(\mathcal{A}(S)) - R_{S_k}(\mathcal{A}_k(S_k)) \right]$ indicates the level of non-iidness for client $k$ in function $\mathcal{A}$ on dataset $S$.*

*Proof.* We again consider $S_k' = (z_{k,1}', ..., z_{k,n_k}')$, where $z_{k,i}'$ is sampled from $\mathcal{D}_k$. Let also define $S^{(k)} = (S_1, ..., S_k', ..., S_K)$. Based on Lemma A.1, we can express the expected generalization error as

$$\mathbb{E}_{\{S_k \sim \mathcal{D}_k^{n_k}\}_{k=1}^K} \Delta_{\mathcal{A}}(S) = \mathbb{E}_{k \sim \mathcal{K}, \{S_k, S_k' \sim \mathcal{D}_k^{n_k}\}_{k=1}^K} \left[ \frac{1}{n_k} \sum_{i=1}^{n_k} \left( l(\mathcal{A}(S), z_{k,i}') - l(\mathcal{A}(S^{(k)}), z_{k,i}') \right) \right]. \tag{16}$$

Based on $L$-smoothness of $l(M_{\boldsymbol{\theta}}, z)$ in $M_{\boldsymbol{\theta}}$, we obtain

$$\frac{1}{n_k} \sum_{i=1}^{n_k} \left( l(\mathcal{A}(S), z_{k,i}') - l(\mathcal{A}(S^{(k)}), z_{k,i}') \right) \le \langle \nabla \frac{1}{n_k} \sum_{i=1}^{n_k} l(\mathcal{A}(S^{(k)}), z_{k,i}'), \mathcal{A}(S) - \mathcal{A}(S^{(k)}) \rangle + \frac{L}{2} \|\mathcal{A}(S) - \mathcal{A}(S^{(k)})\|^2,$$

$$\tag{17}$$

where $\langle \cdot, \cdot \rangle, \| \cdot \|^2$ indicate Euclidean inner product, and squared $L2$-norm. Note that (17) holds due to

$$f(\boldsymbol{y}) \leq f(\boldsymbol{x}) + \langle \nabla f(\boldsymbol{x}), \boldsymbol{y} - \boldsymbol{x} \rangle + \frac{L}{2} \| \boldsymbol{y} - \boldsymbol{x} \|^2. \tag{18}$$

We can bound expectation of the inner product term on the right-hand side of (17) using Cauchy–Schwarz inequality as

$$\mathbb{E}_{k \sim \mathcal{K}, \{\boldsymbol{S}_k, \boldsymbol{S}_k' \sim \mathcal{D}_k^{n_k}\}_{k=1}^K} \langle \nabla \frac{1}{n_k} \sum_{i=1}^{n_k} l(\mathcal{A}(\boldsymbol{S}^{(k)}), \boldsymbol{z}_{k,i}'), \mathcal{A}(\boldsymbol{S}) - \mathcal{A}(\boldsymbol{S}^{(k)}) \rangle$$

$$\leq \mathbb{E}_{k \sim \mathcal{K}} \mathbb{E}_{\{\boldsymbol{S}_k, \boldsymbol{S}_k' \sim \mathcal{D}_k^{n_k}\}_{k=1}^K} |\langle \nabla \frac{1}{n_k} \sum_{i=1}^{n_k} l(\mathcal{A}(\boldsymbol{S}^{(k)}), \boldsymbol{z}_{k,i}'), \mathcal{A}(\boldsymbol{S}) - \mathcal{A}(\boldsymbol{S}^{(k)}) \rangle| \tag{19}$$

$$\leq \mathbb{E}_{k \sim \mathcal{K}} \left[ \mathbb{E}_{\{\boldsymbol{S}_k, \boldsymbol{S}_k' \sim \mathcal{D}_k^{n_k}\}_{k=1}^K} \| \nabla \frac{1}{n_k} \sum_{i=1}^{n_k} l(\mathcal{A}(\boldsymbol{S}^{(k)}), \boldsymbol{z}_{k,i}') \| \quad \| \mathcal{A}(\boldsymbol{S}) - \mathcal{A}(\boldsymbol{S}^{(k)}) \| \right] \tag{20}$$

$$\leq \mathbb{E}_{k \sim \mathcal{K}} \sqrt{\mathbb{E}_{\{\boldsymbol{S}_k, \boldsymbol{S}_k' \sim \mathcal{D}_k^{n_k}\}_{k=1}^K} \| \nabla \frac{1}{n_k} \sum_{i=1}^{n_k} l(\mathcal{A}(\boldsymbol{S}^{(k)}), \boldsymbol{z}_{k,i}') \|^2 \mathbb{E}_{\{\boldsymbol{S}_k, \boldsymbol{S}_k' \sim \mathcal{D}_k^{n_k}\}_{k=1}^K} \| \mathcal{A}(\boldsymbol{S}) - \mathcal{A}(\boldsymbol{S}^{(k)}) \|^2}, \tag{21}$$

where (19) is true because on the right we have the absolute value. (20), and (21) are based on Cauchy–Schwarz inequality. Now Let's find an upper bound for $\mathbb{E}_{\{\boldsymbol{S}_k, \boldsymbol{S}_k' \sim \mathcal{D}_k^{n_k}\}_{k=1}^K} \| \mathcal{A}(\boldsymbol{S}) - \mathcal{A}(\boldsymbol{S}^{(k)}) \|^2$ that appears on the right-hand side of both (17), and (21). We obtain

$$\mathbb{E}_{\{\boldsymbol{S}_k, \boldsymbol{S}_k' \sim \mathcal{D}_k^{n_k}\}_{k=1}^K} \| \mathcal{A}(\boldsymbol{S}) - \mathcal{A}(\boldsymbol{S}^{(k)}) \|^2$$

$$= \mathbb{E}_{\{\boldsymbol{S}_k, \boldsymbol{S}_k' \sim \mathcal{D}_k^{n_k}\}_{k=1}^K} \mathcal{K}(k)^2 \| \mathcal{A}_k(\boldsymbol{S}_k) - \mathcal{A}_k(\boldsymbol{S}_k') \|^2 \tag{22}$$

$$\leq \mathbb{E}_{\{\boldsymbol{S}_k, \boldsymbol{S}_k' \sim \mathcal{D}_k^{n_k}\}_{k=1}^K} \frac{2\mathcal{K}(k)^2}{\mu} \left( R_{\boldsymbol{S}_k'}(\mathcal{A}_k(\boldsymbol{S}_k)) - R_{\boldsymbol{S}_k'}(\mathcal{A}_k(\boldsymbol{S}_k')) \right) \tag{23}$$

$$= \mathbb{E}_{\{\boldsymbol{S}_k, \boldsymbol{S}_k' \sim \mathcal{D}_k^{n_k}\}_{k=1}^K} \frac{2\mathcal{K}(k)^2}{\mu} \frac{1}{n_k} \sum_{j=1}^{n_k} \left( l(\mathcal{A}_k(\boldsymbol{S}_k), \boldsymbol{z}_{k,j}') - l(\mathcal{A}_k(\boldsymbol{S}_k'), \boldsymbol{z}_{k,j}') \right) \tag{24}$$

$$= \mathbb{E}_{\{\boldsymbol{S}_k, \boldsymbol{S}_k' \sim \mathcal{D}_k^{n_k}\}_{k=1}^K} \frac{2\mathcal{K}(k)^2}{\mu} \Delta_{\mathcal{A}_k}(\boldsymbol{S}_k') \tag{25}$$

$$= \mathbb{E}_{\{\boldsymbol{S}_k \sim \mathcal{D}_k^{n_k}\}_{k=1}^K} \frac{2\mathcal{K}(k)^2}{\mu} \Delta_{\mathcal{A}_k}(\boldsymbol{S}_k), \tag{26}$$

where (22) proceeds by observing that $\mathcal{A}(\boldsymbol{S}^{(k,i)})$ varies solely in the sub-model derived from node $k$, diverging from $\mathcal{A}(\boldsymbol{S})$, and this discrepancy is magnified by a factor of $\mathcal{K}(k)$ when averaging of all sub-models. (23) holds due to the $\mu$-strongly convexity of $l(M_{\boldsymbol{\theta}}, \boldsymbol{z})$ in $M_{\boldsymbol{\theta}}$ which leads to $\mu$-strongly convexity of $R_{\boldsymbol{S}_k}(M_{\boldsymbol{\theta}})$ and the fact that $\mathcal{A}_k(\boldsymbol{S}_k')$ is derived from the local ERM, i.e., $\mathcal{A}_k(\boldsymbol{S}_k') = \arg\min_M \left( \sum_{i=1}^{n_k} l(M, \boldsymbol{z}_{k,i}') \right)$ and $\nabla R_{\boldsymbol{S}_k'}(\mathcal{A}_k(\boldsymbol{S}_k')) = 0$. Note that if $f$ is $\mu$-strongly convex, we get

$$f(\boldsymbol{x}) - f(\boldsymbol{y}) + \frac{\mu}{2} \| \boldsymbol{x} - \boldsymbol{y} \|^2 \leq \langle \nabla f(\boldsymbol{x}), \boldsymbol{x} - \boldsymbol{y} \rangle. \tag{27}$$

(24), (25) are based on local empirical and population risk definitions.

Now we bound $\mathbb{E}_{k \sim \mathcal{K}, \{\boldsymbol{S}_k, \boldsymbol{S}_k' \sim \mathcal{D}_k^{n_k}\}_{k=1}^K} \| \nabla \frac{1}{n_k} \sum_{i=1}^{n_k} l(\mathcal{A}(\boldsymbol{S}^{(k)}), \boldsymbol{z}_{k,i}') \|^2$ on the right-hand side of (21). Note that

$$\| \nabla \frac{1}{n_k} \sum_{i=1}^{n_k} l(\mathcal{A}(\boldsymbol{S}^{(k)}), \boldsymbol{z}_{k,i}') \|^2 \leq 2L \left( \frac{1}{n_k} \sum_{i=1}^{n_k} l(\mathcal{A}(\boldsymbol{S}^{(k)}), \boldsymbol{z}_{k,i}') - \frac{1}{n_k} \sum_{i=1}^{n_k} l(\mathcal{A}_k(\boldsymbol{S}_k'), \boldsymbol{z}_{k,i}') \right) \tag{28}$$

$$\leq 2L \left( R_{\boldsymbol{S}_k'}(\mathcal{A}(\boldsymbol{S}^{(k)})) - R_{\boldsymbol{S}_k'}(\mathcal{A}_k(\boldsymbol{S}_k')) \right), \tag{29}$$

where 28 is obtained using the fact that for any $L$-smooth function $f$, we have

$$\|\nabla f(\boldsymbol{x})\|^2 \leq 2L(f(\boldsymbol{x}) - f^*), \tag{30}$$

and the fact that $\mathcal{A}_k(\boldsymbol{S}'_k)$ is derived from the local ERM, *i.e.*, $\mathcal{A}_k(\boldsymbol{S}'_k) = \arg\min_M \sum_{i=1}^{n_k} l(M, \boldsymbol{z}'_{k,i})$. 29 is based on the definition of local empirical risk.

Putting (17) into (16) and considering (21) we get

$$\mathbb{E}_{\{\boldsymbol{S}_k \sim \mathcal{D}_k^{n_k}\}_{k=1}^K} \Delta_{\mathcal{A}}(\boldsymbol{S})$$

$$\leq \mathbb{E}_{k \sim \mathcal{K}} \Bigg[ \mathbb{E}_{\{\boldsymbol{S}_k, \boldsymbol{S}'_k \sim \mathcal{D}_k^{n_k}\}_{k=1}^K} \frac{L}{2} \|\mathcal{A}(\boldsymbol{S}) - \mathcal{A}(\boldsymbol{S}^{(k)})\|^2 \tag{31}$$

$$+ \sqrt{\mathbb{E}_{\{\boldsymbol{S}_k, \boldsymbol{S}'_k \sim \mathcal{D}_k^{n_k}\}_{k=1}^K} \|\nabla \frac{1}{n_k} \sum_{i=1}^{n_k} l(\mathcal{A}(\boldsymbol{S}^{(k)}), \boldsymbol{z}'_{k,i})\|^2 \, \mathbb{E}_{\{\boldsymbol{S}_k, \boldsymbol{S}'_k \sim \mathcal{D}_k^{n_k}\}_{k=1}^K} \|\mathcal{A}(\boldsymbol{S}) - \mathcal{A}(\boldsymbol{S}^{(k)})\|^2} \Bigg]$$

$$\leq \mathbb{E}_{k \sim \mathcal{K}} \Bigg[ \mathbb{E}_{\{\boldsymbol{S}_k \sim \mathcal{D}_k^{n_k}\}_{k=1}^K} \frac{L\mathcal{K}(k)^2}{\mu} \Delta_{\mathcal{A}_k}(\boldsymbol{S}_k) \tag{32}$$

$$+ \sqrt{\mathbb{E}_{\{\boldsymbol{S}_k, \boldsymbol{S}'_k \sim \mathcal{D}_k^{n_k}\}_{k=1}^K} 2L \Big( R_{\boldsymbol{S}'_k}(\mathcal{A}(\boldsymbol{S}^{(k)})) - R_{\boldsymbol{S}'_k}(\mathcal{A}_k(\boldsymbol{S}'_k)) \Big) \, \mathbb{E}_{\{\boldsymbol{S}_k \sim \mathcal{D}_k^{n_k}\}_{k=1}^K} \frac{2\mathcal{K}(k)^2}{\mu} \Delta_{\mathcal{A}_k}(\boldsymbol{S}_k)} \Bigg]$$

$$\leq \mathbb{E}_{k \sim \mathcal{K}} \Bigg[ \mathbb{E}_{\{\boldsymbol{S}_k \sim \mathcal{D}_k^{n_k}\}_{k=1}^K} \frac{L\mathcal{K}(k)^2}{\mu} \Delta_{\mathcal{A}_k}(\boldsymbol{S}_k) \tag{33}$$

$$+ \sqrt{\mathbb{E}_{\{\boldsymbol{S}_k \sim \mathcal{D}_k^{n_k}\}_{k=1}^K} 2L \Big( R_{\boldsymbol{S}_k}(\mathcal{A}(\boldsymbol{S})) - R_{\boldsymbol{S}_k}(\mathcal{A}_k(\boldsymbol{S}_k)) \Big) \, \mathbb{E}_{\{\boldsymbol{S}_k \sim \mathcal{D}_k^{n_k}\}_{k=1}^K} \frac{2\mathcal{K}(k)^2}{\mu} \Delta_{\mathcal{A}_k}(\boldsymbol{S}_k)} \Bigg]$$

$$\leq \mathbb{E}_{k \sim \mathcal{K}} \Bigg[ \frac{L\mathcal{K}(k)^2}{\mu} \mathbb{E}_{\{\boldsymbol{S}_k \sim \mathcal{D}_k^{n_k}\}_{k=1}^K} \Delta_{\mathcal{A}_k}(\boldsymbol{S}_k) + 2\sqrt{\frac{L}{\mu}} \mathcal{K}(k) \sqrt{\mathbb{E}_{\{\boldsymbol{S}_k \sim \mathcal{D}_k^{n_k}\}_{k=1}^K} \delta_{k,\mathcal{A}}(\boldsymbol{S}) \, \mathbb{E}_{\{\boldsymbol{S}_k \sim \mathcal{D}_k^{n_k}\}_{k=1}^K} \Delta_{\mathcal{A}_k}(\boldsymbol{S}_k)} \Bigg] \tag{34}$$

$$, \tag{35}$$

where in (32) we have applied (26), and (29). (33) proceeds by considering that

$$\mathbb{E}_{\{\boldsymbol{S}_k, \boldsymbol{S}'_k \sim \mathcal{D}_k^{n_k}\}_{k=1}^K} \Big[ R_{\boldsymbol{S}'_k}(\mathcal{A}(\boldsymbol{S}^{(k)})) - R_{\boldsymbol{S}'_k}(\mathcal{A}_k(\boldsymbol{S}'_k)) \Big] = \mathbb{E}_{\{\boldsymbol{S}_k \sim \mathcal{D}_k^{n_k}\}_{k=1}^K} \Big[ R_{\boldsymbol{S}_k}(\mathcal{A}(\boldsymbol{S})) - R_{\boldsymbol{S}_k}(\mathcal{A}_k(\boldsymbol{S}_k)) \Big]. \tag{36}$$

In (34) we have used the definition of $\delta_{k,\mathcal{A}}(\boldsymbol{S}) = \Big[ R_{\boldsymbol{S}_k}(\mathcal{A}(\boldsymbol{S})) - R_{\boldsymbol{S}_k}(\mathcal{A}_k(\boldsymbol{S}_k)) \Big]$. This completes the proof and provides the following upper bound for $\mathbb{E}_{\{\boldsymbol{S}_k \sim \mathcal{D}_k^{n_k}\}_{k=1}^K} \Delta_{\mathcal{A}}(\boldsymbol{S})$,

$$\mathbb{E}_{k \sim \mathcal{K}} \Bigg[ \frac{L\mathcal{K}(k)^2}{\mu} \mathbb{E}_{\{\boldsymbol{S}_k \sim \mathcal{D}_k^{n_k}\}} \Delta_{\mathcal{A}_k}(\boldsymbol{S}_k) + 2\sqrt{\frac{L}{\mu}} \mathcal{K}(k) \sqrt{\mathbb{E}_{\{\boldsymbol{S}_k \sim \mathcal{D}_k^{n_k}\}_{k=1}^K} \delta_{k,\mathcal{A}}(\boldsymbol{S}) \, \mathbb{E}_{\{\boldsymbol{S}_k \sim \mathcal{D}_k^{n_k}\}} \Delta_{\mathcal{A}_k}(\boldsymbol{S}_k)} \Bigg]. \tag{37}$$

$\square$

# B. Proof of Theorem 4.3

Here we provide an identical theorem as Theorem A.2, except that instead of ERM, multiple local stochastic gradient descent steps are used as the local optimizer.

**Theorem B.1.** *Let $l(M_{\boldsymbol{\theta}}, \boldsymbol{z})$ be $\mu$-strongly convex and $L$-smooth in $M_{\boldsymbol{\theta}}$, $M_{\boldsymbol{\theta}_k} = \mathcal{A}_k(\boldsymbol{S}_k)$ represents the model obtained by doing multiple local steps as in (5) on local dataset $\boldsymbol{S}_k$, and $M_{\hat{\boldsymbol{\theta}}} = \mathcal{A}(\boldsymbol{S})$ is the model after one round of FedAvg*

$(\hat{\boldsymbol{\theta}} = \mathbb{E}_{k \sim \mathcal{K}} \, \boldsymbol{\theta}_k)$. *Then, the expected generalization error, $\mathbb{E}_{\{\boldsymbol{S}_k \sim \mathcal{D}_k^{n_k}\}_{k=1}^K} \Delta_{\mathcal{A}}(\boldsymbol{S})$, is bounded by*

$$\mathbb{E}_{k \sim \mathcal{K}} \left[ \mathbb{E}_{\{\boldsymbol{S}_k \sim \mathcal{D}_k^{n_k}\}_{k=1}^K} \frac{2LK(k)^2}{\mu} \left( \Delta_{\mathcal{A}_k}(\boldsymbol{S}_k) + 2\epsilon_k(\boldsymbol{S}_k) \right) \right. \tag{38}$$

$$\left. + \sqrt{\frac{8L}{\mu}} \mathcal{K}(k) \sqrt{\mathbb{E}_{\{\boldsymbol{S}_k \sim \mathcal{D}_k^{n_k}\}_{k=1}^K} \left( \delta_{k,\mathcal{A}}(\boldsymbol{S}) + \epsilon_k(\boldsymbol{S}_k) \right) \mathbb{E}_{\{\boldsymbol{S}_k \sim \mathcal{D}_k^{n_k}\}_{k=1}^K} \left( \Delta_{\mathcal{A}_k}(\boldsymbol{S}_k) + 2\epsilon_k(\boldsymbol{S}_k) \right)} \right],$$

*where $\epsilon_k(\boldsymbol{S}_k) = R_{\boldsymbol{S}_k}(\mathcal{A}_k(\boldsymbol{S}_k)) - R_{\boldsymbol{S}_k}(\mathcal{A}^*(\boldsymbol{S}_k))$.*

*Proof.* All the steps are exactly the same as in the proof of theorem A.2 except for the two steps below:

First, we use a new upper bound for $\mathbb{E}_{\{\boldsymbol{S}_k, \boldsymbol{S}_k' \sim \mathcal{D}_k^{n_k}\}_{k=1}^K} \|\mathcal{A}(\boldsymbol{S}) - \mathcal{A}(\boldsymbol{S}^{(k)})\|^2$ that appears on the right-hand side of both (17), and (21). We have

$$\mathbb{E}_{\{\boldsymbol{S}_k, \boldsymbol{S}_k' \sim \mathcal{D}_k^{n_k}\}_{k=1}^K} \|\mathcal{A}(\boldsymbol{S}) - \mathcal{A}(\boldsymbol{S}^{(k)})\|^2$$

$$= \mathbb{E}_{\{\boldsymbol{S}_k, \boldsymbol{S}_k' \sim \mathcal{D}_k^{n_k}\}_{k=1}^K} \mathcal{K}(k)^2 \|\mathcal{A}_k(\boldsymbol{S}_k) - \mathcal{A}_k(\boldsymbol{S}_k')\|^2 \tag{39}$$

$$= \mathbb{E}_{\{\boldsymbol{S}_k, \boldsymbol{S}_k' \sim \mathcal{D}_k^{n_k}\}_{k=1}^K} \mathcal{K}(k)^2 \|\mathcal{A}_k(\boldsymbol{S}_k) - \mathcal{A}^*(\boldsymbol{S}_k') + \mathcal{A}^*(\boldsymbol{S}_k') - \mathcal{A}_k(\boldsymbol{S}_k')\|^2 \tag{40}$$

$$= \mathbb{E}_{\{\boldsymbol{S}_k, \boldsymbol{S}_k' \sim \mathcal{D}_k^{n_k}\}_{k=1}^K} 2\mathcal{K}(k)^2 \left( \|\mathcal{A}_k(\boldsymbol{S}_k) - \mathcal{A}^*(\boldsymbol{S}_k')\|^2 + \|\mathcal{A}^*(\boldsymbol{S}_k') - \mathcal{A}_k(\boldsymbol{S}_k')\|^2 \right) \tag{41}$$

$$\leq \mathbb{E}_{\{\boldsymbol{S}_k, \boldsymbol{S}_k' \sim \mathcal{D}_k^{n_k}\}_{k=1}^K} \frac{4\mathcal{K}(k)^2}{\mu} \left( R_{\boldsymbol{S}_k'}(\mathcal{A}_k(\boldsymbol{S}_k)) - R_{\boldsymbol{S}_k'}(\mathcal{A}^*(\boldsymbol{S}_k')) + R_{\boldsymbol{S}_k'}(\mathcal{A}_k(\boldsymbol{S}_k')) - R_{\boldsymbol{S}_k'}(\mathcal{A}^*(\boldsymbol{S}_k')) \right) \tag{42}$$

$$= \mathbb{E}_{\{\boldsymbol{S}_k, \boldsymbol{S}_k' \sim \mathcal{D}_k^{n_k}\}_{k=1}^K} \frac{4\mathcal{K}(k)^2}{\mu} \left( R_{\boldsymbol{S}_k'}(\mathcal{A}_k(\boldsymbol{S}_k)) - R_{\boldsymbol{S}_k'}(\mathcal{A}_k(\boldsymbol{S}_k')) + 2R_{\boldsymbol{S}_k'}(\mathcal{A}_k(\boldsymbol{S}_k')) - 2R_{\boldsymbol{S}_k'}(\mathcal{A}^*(\boldsymbol{S}_k')) \right) \tag{43}$$

$$= \mathbb{E}_{\{\boldsymbol{S}_k, \boldsymbol{S}_k' \sim \mathcal{D}_k^{n_k}\}_{k=1}^K} \frac{4\mathcal{K}(k)^2}{\mu} \left( \frac{1}{n_k} \sum_{j=1}^{n_k} \left( l(\mathcal{A}_k(\boldsymbol{S}_k), z_{k,j}') - l(\mathcal{A}_k(\boldsymbol{S}_k'), z_{k,j}') \right) + 2R_{\boldsymbol{S}_k'}(\mathcal{A}_k(\boldsymbol{S}_k')) - 2R_{\boldsymbol{S}_k'}(\mathcal{A}^*(\boldsymbol{S}_k')) \right) \tag{44}$$

$$= \mathbb{E}_{\{\boldsymbol{S}_k, \boldsymbol{S}_k' \sim \mathcal{D}_k^{n_k}\}_{k=1}^K} \frac{4\mathcal{K}(k)^2}{\mu} \left( \Delta_{\mathcal{A}_k}(\boldsymbol{S}_k') + 2R_{\boldsymbol{S}_k'}(\mathcal{A}_k(\boldsymbol{S}_k')) - 2R_{\boldsymbol{S}_k'}(\mathcal{A}^*(\boldsymbol{S}_k')) \right) \tag{45}$$

$$= \mathbb{E}_{\{\boldsymbol{S}_k \sim \mathcal{D}_k^{n_k}\}_{k=1}^K} \frac{4\mathcal{K}(k)^2}{\mu} \left( \Delta_{\mathcal{A}_k}(\boldsymbol{S}_k) + 2R_{\boldsymbol{S}_k}(\mathcal{A}_k(\boldsymbol{S}_k)) - 2R_{\boldsymbol{S}_k}(\mathcal{A}^*(\boldsymbol{S}_k)) \right), \tag{46}$$

where in (40) $\mathcal{A}^*(\boldsymbol{S}_k)$ is the ERM on $\boldsymbol{S}_k$. (41) is based on the following inequality.

$$\| \sum_{i=1}^{n} a_i \|^2 \leq n \sum_{i=1}^{n} \|a_i\|^2. \tag{47}$$

Secondly, we derive a new bound for $\mathbb{E}_{k \sim \mathcal{K}, \{\boldsymbol{S}_k, \boldsymbol{S}_k' \sim \mathcal{D}_k^{n_k}\}_{k=1}^K} \|\nabla \frac{1}{n_k} \sum_{i=1}^{n_k} l(\mathcal{A}(\boldsymbol{S}^{(k)}), z_{k,i}')\|^2$ on the right hand side of (21). We get

$$\|\nabla \frac{1}{n_k} \sum_{i=1}^{n_k} l(\mathcal{A}(\boldsymbol{S}^{(k)}), z_{k,i}')\|^2 \leq 2L \left( \frac{1}{n_k} \sum_{i=1}^{n_k} l(\mathcal{A}(\boldsymbol{S}^{(k)}), z_{k,i}') - \frac{1}{n_k} \sum_{i=1}^{n_k} l(\mathcal{A}^*(\boldsymbol{S}_k'), z_{k,i}') \right) \tag{48}$$

$$\leq 2L \left( R_{\boldsymbol{S}_k'}(\mathcal{A}(\boldsymbol{S}^{(k)})) - R_{\boldsymbol{S}_k'}(\mathcal{A}^*(\boldsymbol{S}_k')) \right) \tag{49}$$

$$\leq 2L \left( R_{\boldsymbol{S}_k'}(\mathcal{A}(\boldsymbol{S}^{(k)})) - R_{\boldsymbol{S}_k'}(\mathcal{A}_k(\boldsymbol{S}_k')) + R_{\boldsymbol{S}_k'}(\mathcal{A}_k(\boldsymbol{S}_k')) - R_{\boldsymbol{S}_k'}(\mathcal{A}^*(\boldsymbol{S}_k')) \right) \tag{50}$$

$$\leq 2L \left( \delta_{k,\mathcal{A}}(\boldsymbol{S}^{(k)}) + R_{\boldsymbol{S}_k'}(\mathcal{A}_k(\boldsymbol{S}_k')) - R_{\boldsymbol{S}_k'}(\mathcal{A}^*(\boldsymbol{S}_k')) \right). \tag{51}$$

Let's define $\epsilon_k(\boldsymbol{S}_k) = R_{\boldsymbol{S}_k}(\mathcal{A}_k(\boldsymbol{S}_k)) - R_{\boldsymbol{S}_k}(\mathcal{A}^*(\boldsymbol{S}_k))$. Putting (17) into (16) and considering (21) we get

$$
\begin{aligned}
\mathbb{E}_{\{\boldsymbol{S}_k \sim \mathcal{D}_k^{n_k}\}_{k=1}^K} \Delta_{\mathcal{A}}(\boldsymbol{S}) &\leq \mathbb{E}_{k \sim \mathcal{K}} \Bigg[ \mathbb{E}_{\{\boldsymbol{S}_k, \boldsymbol{S}_k' \sim \mathcal{D}_k^{n_k}\}_{k=1}^K} \frac{L}{2} \| \mathcal{A}(\boldsymbol{S}) - \mathcal{A}(\boldsymbol{S}^{(k)}) \|^2 \\
&\quad + \sqrt{\mathbb{E}_{\{\boldsymbol{S}_k, \boldsymbol{S}_k' \sim \mathcal{D}_k^{n_k}\}_{k=1}^K} \| \nabla \frac{1}{n_k} \sum_{i=1}^{n_k} l(\mathcal{A}(\boldsymbol{S}^{(k)}), \boldsymbol{z}_{k,i}') \|^2 \, \mathbb{E}_{\{\boldsymbol{S}_k, \boldsymbol{S}_k' \sim \mathcal{D}_k^{n_k}\}_{k=1}^K} \| \mathcal{A}(\boldsymbol{S}) - \mathcal{A}(\boldsymbol{S}^{(k)}) \|^2} \Bigg]
\end{aligned}
$$

$$
\leq \mathbb{E}_{k \sim \mathcal{K}} \Bigg[ \mathbb{E}_{\{\boldsymbol{S}_k \sim \mathcal{D}_k^{n_k}\}_{k=1}^K} \frac{2L\mathcal{K}(k)^2}{\mu} \Big( \Delta_{\mathcal{A}_k}(\boldsymbol{S}_k) + 2\epsilon_k(\boldsymbol{S}_k) \Big) \tag{52}
$$

$$
+ \sqrt{\mathbb{E}_{\{\boldsymbol{S}_k, \boldsymbol{S}_k' \sim \mathcal{D}_k^{n_k}\}_{k=1}^K} 2L\Big(\delta_{k, \mathcal{A}}(\boldsymbol{S}^{(k)}) + \epsilon_k(\boldsymbol{S}_k')\Big) \mathbb{E}_{\{\boldsymbol{S}_k \sim \mathcal{D}_k^{n_k}\}_{k=1}^K} \frac{4\mathcal{K}(k)^2}{\mu} \Big( \Delta_{\mathcal{A}_k}(\boldsymbol{S}_k) + 2\epsilon_k(\boldsymbol{S}_k) \Big)} \Bigg]
$$

$$
\leq \mathbb{E}_{k \sim \mathcal{K}} \Bigg[ \mathbb{E}_{\{\boldsymbol{S}_k \sim \mathcal{D}_k^{n_k}\}_{k=1}^K} \frac{2L\mathcal{K}(k)^2}{\mu} \Big( \Delta_{\mathcal{A}_k}(\boldsymbol{S}_k) + 2\epsilon_k(\boldsymbol{S}_k) \Big) \tag{53}
$$

$$
+ \sqrt{\frac{8L}{\mu}} \mathcal{K}(k) \sqrt{\mathbb{E}_{\{\boldsymbol{S}_k \sim \mathcal{D}_k^{n_k}\}_{k=1}^K} \Big( \delta_{k, \mathcal{A}}(\boldsymbol{S}) + \epsilon_k(\boldsymbol{S}_k) \Big) \mathbb{E}_{\{\boldsymbol{S}_k \sim \mathcal{D}_k^{n_k}\}_{k=1}^K} \Big( \Delta_{\mathcal{A}_k}(\boldsymbol{S}_k) + 2\epsilon_k(\boldsymbol{S}_k) \Big)} \Bigg],
$$

where in (52) we have applied (46), and (51). This completes the proof. $\qquad\square$

Now, we prove Theorem 4.3 as follows.

**Theorem B.2.** *Let $l(M_{\boldsymbol{\theta}}, \boldsymbol{z})$ be $\mu$-strongly convex and $L$-smooth in $M_{\boldsymbol{\theta}}$. Local models at round $r$ are calculated by doing $\tau$ local gradient descent steps (5), and the local gradient variance is bounded by $\sigma^2$, i.e., $\mathbb{E}_{\boldsymbol{z} \sim \mathcal{D}_k} \|\nabla l(M_{\boldsymbol{\theta}}, \boldsymbol{z}) - \mathbb{E}_{\boldsymbol{z} \sim \mathcal{D}_k} \nabla l(M_{\boldsymbol{\theta}}, \boldsymbol{z})\|^2 \leq \sigma^2$. The aggregated model at round $r$, $M_{\hat{\boldsymbol{\theta}}_r}$, is obtained by performing FedAvg, and the data points used in round $r$ (i.e., $Z_{k,r}$) are sampled without replacement. The average generalization error bound is*

$$
\mathbb{E}_{\boldsymbol{S}} \Delta_{FedAvg}(\boldsymbol{S}) \leq \frac{1}{R} \sum_{r=1}^R \mathbb{E}_{k \sim \mathcal{K}} \left[ \frac{2L\mathcal{K}(k)^2}{\mu} A + \sqrt{\frac{8L}{\mu}} \mathcal{K}(k)(AB)^{\frac{1}{2}} \right]
$$

*where $A = \tilde{O}\left( \sqrt{\frac{\mathcal{C}(M_{\boldsymbol{\theta}})}{|Z_{k,r}|}} + \frac{\sigma^2}{\mu\tau} + \frac{L}{\mu} \right)$, $B = \tilde{O}\left( \mathbb{E}_{\{Z_{k,r}\}_{k=1}^K} \delta_{k, \mathcal{A}}(\{Z_{k,r}\}_{k=1}^K) + \frac{\sigma^2}{\mu\tau} + \frac{L}{\mu} \right)$, $\tilde{O}$ hides constants and poly-logarithmic factors, and $\mathcal{C}(M_{\boldsymbol{\theta}})$ shows the complexity of the model class of $M_{\boldsymbol{\theta}}$.*

*Proof.* Based on the definition, we have

$$\mathbb{E}_{\{\boldsymbol{S}_k \sim \mathcal{D}_k^{n_k}\}_{k=1}^K} \Delta_{FedAvg}(\boldsymbol{S}) = \mathbb{E}_{\{\boldsymbol{S}_k \sim \mathcal{D}_k^{n_k}\}_{k=1}^K} \frac{1}{R} \sum_{r=1}^R \mathbb{E}_{k \sim \mathcal{K}} \left[ \mathbb{E}_{\boldsymbol{z} \sim \mathcal{D}_k} l(M_{\hat{\boldsymbol{\theta}}_r}, \boldsymbol{z}) - \frac{1}{|Z_{k,r}|} \sum_{i \in Z_{k,r}}^K l(M_{\hat{\boldsymbol{\theta}}_r}, \boldsymbol{z}_{k,i}) \right] \quad (54)$$

$$= \frac{1}{R} \sum_{r=1}^R \mathbb{E}_{\{Z_{k,r} \sim \mathcal{D}_k^{|Z_{k,r}|}\}_{k=1}^K} \mathbb{E}_{k \sim \mathcal{K}} \left[ \mathbb{E}_{\boldsymbol{z} \sim \mathcal{D}_k} l(M_{\hat{\boldsymbol{\theta}}_r}, \boldsymbol{z}) - \frac{1}{|Z_{k,r}|} \sum_{i \in Z_{k,r}}^K l(M_{\hat{\boldsymbol{\theta}}_r}, \boldsymbol{z}_{k,i}) \right] \quad (55)$$

$$= \frac{1}{R} \sum_{r=1}^R \mathbb{E}_{\{Z_{k,r} \sim \mathcal{D}_k^{|Z_{k,r}|}\}_{k=1}^K} \Delta_{\mathcal{A}}(\{Z_{k,r}\}_{k=1}^K) \quad (56)$$

$$\leq \frac{1}{R} \sum_{r=1}^R \mathbb{E}_{k \sim \mathcal{K}} \left[ \mathbb{E}_{\{Z_{k,r} \sim \mathcal{D}_k^{|Z_{k,r}|}\}} \frac{2LK(k)^2}{\mu} \left( \Delta_{\mathcal{A}_k}(Z_{k,r}) + 2\epsilon_k(Z_{k,r}) \right) \right.$$
$$\left. + \sqrt{\frac{8L}{\mu}} \mathcal{K}(k) \sqrt{\mathbb{E}_{\{Z_{k,r} \sim \mathcal{D}_k^{|Z_{k,r}|}\}_{k=1}^K} \left( \delta_{k,\mathcal{A}}(\{Z_{k,r}\}_{k=1}^K) + \epsilon_k(Z_{k,r}) \right) \mathbb{E}_{\{Z_{k,r} \sim \mathcal{D}_k^{|Z_{k,r}|}\}} \left( \Delta_{\mathcal{A}_k}(Z_{k,r}) + \epsilon_k(Z_{k,r}) \right)} \right] \quad (57)$$

$$\leq \frac{1}{R} \sum_{r=1}^R \mathbb{E}_{k \sim \mathcal{K}} \left[ \frac{2LK(k)^2}{\mu} \tilde{O}\left( \sqrt{\frac{\mathcal{C}(M_{\boldsymbol{\theta}})}{|Z_{k,r}|}} + \frac{\sigma^2}{\mu\tau} + \frac{L}{\mu} \right) \right.$$
$$\left. + \sqrt{\frac{8L}{\mu}} \mathcal{K}(k) \sqrt{\tilde{O}\left( \mathbb{E}_{\{Z_{k,r} \sim \mathcal{D}_k^{|Z_{k,r}|}\}_{k=1}^K} \delta_{k,\mathcal{A}}(\{Z_{k,r}\}_{k=1}^K) + \frac{\sigma^2}{\mu\tau} + \frac{L}{\mu} \right) \tilde{O}\left( \sqrt{\frac{\mathcal{C}(M_{\boldsymbol{\theta}})}{|Z_{k,r}|}} + \frac{\sigma^2}{\mu\tau} + \frac{L}{\mu} \right)} \right], \quad (58)$$

where in (56), $\mathcal{A}$ represents one-round FedAvg algorithm. In (57) we have used Theorem B.1. In (58) we have used the conventional statistical learning theory originated with Leslie Valiant's probably approximately correct (PAC) framework (Valiant, 1984). We have also applied the optimization convergence rate bounds in the literature (Stich & Karimireddy, 2019). Note that $\tilde{O}$ hides constants and poly-logarithmic factors. $\qquad \square$

## C. Partial Client Participation Setting

We first define an empirical risk for the partial participation distribution $\hat{\mathcal{K}}$ on dataset $\boldsymbol{S}$, where $Supp(\hat{\mathcal{K}}) \neq \{1, \dots, K\}$ and $|Supp(\hat{\mathcal{K}})| = \hat{K} \leq K$, as

$$R_{\boldsymbol{S}}^{\hat{\mathcal{K}}}(M_{\boldsymbol{\theta}}) = \mathbb{E}_{k \sim \hat{\mathcal{K}}} R_{\boldsymbol{S}_k}(M_{\boldsymbol{\theta}}) = \mathbb{E}_{k \sim \hat{\mathcal{K}}} \frac{1}{n_k} \sum_{i=1}^{n_k} l(M_{\boldsymbol{\theta}}, \boldsymbol{z}_{k,i}), \quad (59)$$

where $\hat{\mathcal{K}}$ is an arbitrary distribution on participating nodes that is a part of all nodes, and $R_{\boldsymbol{S}_k}(M_{\boldsymbol{\theta}})$ is the empirical risk for model $M_{\boldsymbol{\theta}}$ on local dataset $\boldsymbol{S}_k$. We further define a partial population risk for model $M_{\boldsymbol{\theta}}$ as

$$R^{\hat{\mathcal{K}}}(M_{\boldsymbol{\theta}}) = \mathbb{E}_{k \sim \hat{\mathcal{K}}} R_k(M_{\boldsymbol{\theta}}) = \mathbb{E}_{k \sim \hat{\mathcal{K}}, \boldsymbol{z} \sim \mathcal{D}_k} l(M_{\boldsymbol{\theta}}, \boldsymbol{z}), \quad (60)$$

where $R_k(M_{\boldsymbol{\theta}})$ is the population risk on node $k$'s data distribution.

Now, we can define the generalization error for dataset $\boldsymbol{S}$ and function $\mathcal{A}(\boldsymbol{S})$ as

$$\Delta_{\mathcal{A}}(\boldsymbol{S}) = R(\mathcal{A}(\boldsymbol{S})) - R_{\boldsymbol{S}}^{\hat{\mathcal{K}}}(\mathcal{A}(\boldsymbol{S})) \quad (61)$$

$$= \underbrace{R(\mathcal{A}(\boldsymbol{S})) - R^{\hat{\mathcal{K}}}(\mathcal{A}(\boldsymbol{S}))}_{\text{Participation gap}} + \underbrace{R^{\hat{\mathcal{K}}}(\mathcal{A}(\boldsymbol{S})) - R_{\boldsymbol{S}}^{\hat{\mathcal{K}}}(\mathcal{A}(\boldsymbol{S}))}_{\text{Out-of-sample gap}}. \quad (62)$$

The expected generalization error is expressed as $\mathbb{E}_{\boldsymbol{S}_k \sim \mathcal{D}k^{n_k}{}_{k=1}^K} \Delta_{\mathcal{A}}(\boldsymbol{S})$. Note that the second term in (62), which is related to the difference between in-sample and out-of-sample loss, can be bounded in the same way as in Theorem 4.1 and Theorem 4.3. The first term is associated with the participation of not all clients. In the following, we demonstrate that under certain conditions, this term would be zero in expectation.

We assume there is a meta-distribution $\mathcal{P}$ supported on all distributions $\hat{K}$.

**Lemma C.1.** *Let $\{x_i\}_{i=1}^K$ denote any fixed deterministic sequence. Assume $\mathcal{P}$ is derived by sampling $\hat{K}$ clients with replacement based on distribution $\mathcal{K}$ followed by an equal probability on all sampled clients,* i.e., $\hat{\mathcal{K}}(k) = \frac{1}{\hat{K}}$. *Then*

$$\mathbb{E}_{\mathcal{P}}\,\mathbb{E}_{k\sim\hat{\mathcal{K}}}\,x_k = \mathbb{E}_{k\sim\mathcal{K}}\,x_k, \mathbb{E}_{\mathcal{P}}\,\mathbb{E}_{k\sim\hat{\mathcal{K}}}\,\hat{\mathcal{K}}(k)x_k = \frac{1}{\hat{K}}\,\mathbb{E}_{k\sim\mathcal{K}}\,x_k, \mathbb{E}_{\mathcal{P}}\,\mathbb{E}_{k\sim\hat{\mathcal{K}}}\,\hat{\mathcal{K}}^2(k)x_k = \frac{1}{\hat{K}^2}\,\mathbb{E}_{k\sim\mathcal{K}}\,x_k. \tag{63}$$

*Proof.*

$$\mathbb{E}_{\mathcal{P}}\,\mathbb{E}_{k\sim\hat{\mathcal{K}}}\,x_k = \mathbb{E}_{\mathcal{P}}\,\frac{1}{\hat{K}}\sum_{i=1}^{\hat{K}}x_i = \frac{1}{\hat{K}}\sum_{i=1}^{\hat{K}}\mathbb{E}_{\mathcal{P}}\,x_i = \mathbb{E}_{\mathcal{P}}\,x_i = \mathbb{E}_{k\sim\mathcal{K}}\,x_k \tag{64}$$

$$\mathbb{E}_{\mathcal{P}}\,\mathbb{E}_{k\sim\hat{\mathcal{K}}}\,\hat{\mathcal{K}}(k)x_k = \mathbb{E}_{\mathcal{P}}\,\frac{1}{\hat{K}^2}\sum_{i=1}^{\hat{K}}x_i = \frac{1}{\hat{K}^2}\sum_{i=1}^{\hat{K}}\mathbb{E}_{\mathcal{P}}\,x_i = \frac{1}{\hat{K}}\,\mathbb{E}_{\mathcal{P}}\,x_i = \frac{1}{\hat{K}}\,\mathbb{E}_{k\sim\mathcal{K}}\,x_k \tag{65}$$

$$\mathbb{E}_{\mathcal{P}}\,\mathbb{E}_{k\sim\hat{\mathcal{K}}}\,\hat{\mathcal{K}}^2(k)x_k = \mathbb{E}_{\mathcal{P}}\,\frac{1}{\hat{K}^3}\sum_{i=1}^{\hat{K}}x_i = \frac{1}{\hat{K}^3}\sum_{i=1}^{\hat{K}}\mathbb{E}_{\mathcal{P}}\,x_i = \frac{1}{\hat{K}^2}\,\mathbb{E}_{\mathcal{P}}\,x_i = \frac{1}{\hat{K}^2}\,\mathbb{E}_{k\sim\mathcal{K}}\,x_k \tag{66}$$

$\square$

**Lemma C.2.** *Let $\{x_i\}_{i=1}^K$ denote any fixed deterministic sequence. Assume $\mathcal{P}$ is derived by sampling $\hat{K}$ clients without replacement uniformly at random followed by weighted probability on all sampled clients as $\hat{\mathcal{K}}(k) = \frac{\mathcal{K}(k)K}{\hat{K}}$. Then*

$$\mathbb{E}_{\mathcal{P}}\,\mathbb{E}_{k\sim\hat{\mathcal{K}}}\,x_k = \mathbb{E}_{k\sim\mathcal{K}}\,x_k, \mathbb{E}_{\mathcal{P}}\,\mathbb{E}_{k\sim\hat{\mathcal{K}}}\,\hat{\mathcal{K}}(k)x_k = \frac{K}{\hat{K}}\,\mathbb{E}_{k\sim\mathcal{K}}\,\mathcal{K}(k)x_k, \mathbb{E}_{\mathcal{P}}\,\mathbb{E}_{k\sim\hat{\mathcal{K}}}\,\hat{\mathcal{K}}^2(k)x_k = \frac{K^2}{\hat{K}^2}\,\mathbb{E}_{k\sim\mathcal{K}}\,\mathcal{K}^2(k)x_k. \tag{67}$$

*Proof.*

$$\mathbb{E}_{\mathcal{P}}\,\mathbb{E}_{k\sim\hat{\mathcal{K}}}\,x_k = \mathbb{E}_{\mathcal{P}}\,\frac{K}{\hat{K}}\sum_{i=1}^{\hat{K}}\mathcal{K}(i)x_i = \frac{K}{\hat{K}}\sum_{i=1}^{\hat{K}}\mathbb{E}_{\mathcal{P}}\,\mathcal{K}(i)x_i = K\,\mathbb{E}_{\mathcal{P}}\,\mathcal{K}(i)x_i = K\frac{1}{K}\sum_{i=1}^{k}\mathcal{K}(i)x_i = \mathbb{E}_{k\sim\mathcal{K}}\,x_k \tag{68}$$

$$\mathbb{E}_{\mathcal{P}}\,\mathbb{E}_{k\sim\hat{\mathcal{K}}}\,\hat{\mathcal{K}}(k)x_k = \mathbb{E}_{\mathcal{P}}\,\frac{K^2}{\hat{K}^2}\sum_{i=1}^{\hat{K}}\mathcal{K}^2(i)x_i = \frac{K^2}{\hat{K}^2}\sum_{i=1}^{\hat{K}}\mathbb{E}_{\mathcal{P}}\,\mathcal{K}^2(i)x_i = \frac{K^2}{\hat{K}}\,\mathbb{E}_{\mathcal{P}}\,\mathcal{K}^2(i)x_i = \frac{K^2}{\hat{K}}\frac{1}{K}\sum_{i=1}^{k}\mathcal{K}^2(i)x_i \tag{69}$$

$$= \frac{K}{\hat{K}}\,\mathbb{E}_{k\sim\mathcal{K}}\,\mathcal{K}(k)x_k$$

$$\mathbb{E}_{\mathcal{P}}\,\mathbb{E}_{k\sim\hat{\mathcal{K}}}\,\hat{\mathcal{K}}^2(k)x_k = \mathbb{E}_{\mathcal{P}}\,\frac{K^3}{\hat{K}^3}\sum_{i=1}^{\hat{K}}\mathcal{K}^3(i)x_i = \frac{K^3}{\hat{K}^3}\sum_{i=1}^{\hat{K}}\mathbb{E}_{\mathcal{P}}\,\mathcal{K}^3(i)x_i = \frac{K^3}{\hat{K}^2}\,\mathbb{E}_{\mathcal{P}}\,\mathcal{K}^3(i)x_i = \frac{K^3}{\hat{K}^2}\frac{1}{K}\sum_{i=1}^{k}\mathcal{K}^3(i)x_i \tag{70}$$

$$= \frac{K^2}{\hat{K}^2}\,\mathbb{E}_{k\sim\mathcal{K}}\,\mathcal{K}^2(k)x_k$$

$\square$

So based on lemmas C.1, and C.2, it becomes evident that the expectation of the participation gap in (62) becomes zero for both two methods, *i.e.,*

$$\mathbb{E}_{\mathcal{P}}\left[R(\mathcal{A}(\boldsymbol{S})) - R^{\hat{\mathcal{K}}}(\mathcal{A}(\boldsymbol{S}))\right] = 0. \tag{71}$$

The expected generalization error, $\mathbb{E}_{\{\boldsymbol{S}_k\sim\mathcal{D}_k^{n_k}\}_{k=1}^K}\,\Delta_{\mathcal{A}}(\boldsymbol{S})$,will be just the expectation of the second term in 62 that can be bounded using Lemma A.2 by

$$\mathbb{E}_{k\sim\hat{\mathcal{K}}}\left[\frac{L\hat{\mathcal{K}}(k)^2}{\mu}\,\mathbb{E}_{\{\boldsymbol{S}_k\sim\mathcal{D}_k^{n_k}\}}\,\Delta_{\mathcal{A}_k}(\boldsymbol{S}_k) + 2\sqrt{\frac{L}{\mu}}\hat{\mathcal{K}}(k)\sqrt{\mathbb{E}_{\{\boldsymbol{S}_k\sim\mathcal{D}_k^{n_k}\}_{k=1}^K}\,\delta_{k,\mathcal{A}}(\boldsymbol{S})\,\mathbb{E}_{\{\boldsymbol{S}_k\sim\mathcal{D}_k^{n_k}\}}\,\Delta_{\mathcal{A}_k}(\boldsymbol{S}_k)}\right]. \tag{72}$$

---

**Algorithm 2** FedAvg

---

**Input**: Initial model $\{\boldsymbol{\theta}_{k,1,0}\}_{k=1}^K$, Learning rate $\eta$, and number of local steps $\tau$.

**Output**: $\hat{\boldsymbol{\theta}}_R$

1: **for** Round $r$ in $1, ..., R$ **do**
2:     **for** Node $k$ in $1, ..., K$ **in parallel do**
3:         **for** Local step $t$ in $0, ..., \tau-1$ **do**
4:             Sample the batch $\mathcal{B}_{k,r,t}$ from $\mathcal{D}_k$.
5:             $\boldsymbol{\theta}_{k,r,t+1} = \boldsymbol{\theta}_{k,r,t} - \frac{\eta}{|\mathcal{B}_{k,r,t}|} \sum_{i \in \mathcal{B}_{k,r,t}} \nabla l(M_{\boldsymbol{\theta}_{k,r,t}}, \boldsymbol{z}_{k,i})$
6:         $\boldsymbol{\theta}_{k,r+1,0} = \frac{1}{K} \sum_{k=1}^K \boldsymbol{\theta}_{k,r,\tau}$
7: **return** $\hat{\boldsymbol{\theta}}_R = \frac{1}{K} \sum_{k=1}^K \boldsymbol{\theta}_{k,R,\tau}$

---

**Algorithm 3** SCAFFOLD

---

**Input**: Initial model $\{\boldsymbol{\theta}_{k,1,0}\}_{k=1}^K$, Initial control variable $\{\boldsymbol{c}_{k,1}\}_{k=1}^K$, learning rate $\eta$, and number of local steps $\tau$.

**Output**: $\hat{\boldsymbol{\theta}}_R$

1: **for** Round $r$ in $1, ..., R$ **do**
2:     **for** Node $k$ in $1, ..., K$ **in parallel do**
3:         **for** Local step $t$ in $0, ..., \tau-1$ **do**
4:             Sample the batch $\mathcal{B}_{k,r,t}$ from $\mathcal{D}_k$.
5:             $\boldsymbol{\theta}_{k,r,t+1} = \boldsymbol{\theta}_{k,r,t} - \eta\big(\frac{1}{|\mathcal{B}_{k,r,t}|} \sum_{i \in \mathcal{B}_{k,r,t}} \nabla l(M_{\boldsymbol{\theta}_{k,r,t}}, \boldsymbol{z}_{k,i}) - \boldsymbol{c}_{k,r} + \frac{1}{K} \sum_{k=1}^K \boldsymbol{c}_{k,r}\big)$
6:         $\boldsymbol{c}_{k,r+1} = \boldsymbol{c}_{k,r} - \frac{1}{K} \sum_{k=1}^K \boldsymbol{c}_{k,r} + \frac{1}{\eta\tau}(\boldsymbol{\theta}_{k,r,0} - \boldsymbol{\theta}_{k,r,\tau})$
7:         $\boldsymbol{\theta}_{k,r+1,0} = \frac{1}{K} \sum_{k=1}^K \boldsymbol{\theta}_{k,r,\tau}$
8: **return** $\hat{\boldsymbol{\theta}}_R = \frac{1}{K} \sum_{k=1}^K \boldsymbol{\theta}_{k,R,\tau}$

---

If we take the expectation of (72) with respect to $\mathcal{P}$, and taking into account Lemma C.1, we get the generalization bound for method 1 as

$$\mathbb{E}_{k \sim \mathcal{K}} \left[ \frac{L}{\mu \hat{K}^2} \mathbb{E}_{\{\boldsymbol{S}_k \sim \mathcal{D}_k^{n_k}\}} \Delta_{\mathcal{A}_k}(\boldsymbol{S}_k) + 2\sqrt{\frac{L}{\mu} \frac{1}{\hat{K}}} \sqrt{\mathbb{E}_{\{\boldsymbol{S}_k \sim \mathcal{D}_k^{n_k}\}_{k=1}^K} \delta_{k,\mathcal{A}}(\boldsymbol{S}) \mathbb{E}_{\{\boldsymbol{S}_k \sim \mathcal{D}_k^{n_k}\}} \Delta_{\mathcal{A}_k}(\boldsymbol{S}_k)} \right]. \tag{73}$$

For Scheme 2, we can obtain the generalization bound in the same way by taking the expectation of (72) with respect to $\mathcal{P}$ and considering Lemma C.2. We get the generalization bound for Method 2 as:

$$\mathbb{E}_{k \sim \mathcal{K}} \left[ \frac{L}{\mu} \frac{\mathcal{K}(k)^2 K^2}{\hat{K}^2} \mathbb{E}_{\{\boldsymbol{S}_k \sim \mathcal{D}_k^{n_k}\}} \Delta_{\mathcal{A}_k}(\boldsymbol{S}_k) + 2\sqrt{\frac{L}{\mu} \frac{\mathcal{K}(k)K}{\hat{K}}} \sqrt{\mathbb{E}_{\{\boldsymbol{S}_k \sim \mathcal{D}_k^{n_k}\}_{k=1}^K} \delta_{k,\mathcal{A}}(\boldsymbol{S}) \mathbb{E}_{\{\boldsymbol{S}_k \sim \mathcal{D}_k^{n_k}\}} \Delta_{\mathcal{A}_k}(\boldsymbol{S}_k)} \right]. \tag{74}$$

# D. Algorithms Used in the Experiments

In this section, we have listed our implementation of FedAvg (Algorithm 2), SCAFFOLD (Karimireddy et al., 2019) (Algorithm 3), and FedALS+SCAFFOLD (Algorithm 4). In Algorithm 3, we observe that in addition to the model, SCAFFOLD also keeps track of a state-specific to each client, referred to as the client control variate $\boldsymbol{c}_{k,r}$. It is important to recognize that the clients within SCAFFOLD have memory and preserve the $\boldsymbol{c}_{k,r}$ and $\sum_{k=1}^K \boldsymbol{c}_{k,r}$ values. Additionally, when $\boldsymbol{c}_{k,r}$ consistently remains at 0, SCAFFOLD essentially becomes equivalent to FedAvg.

Algorithm 4 demonstrates the integration of FedALS and SCAFFOLD. It is important to observe that in this algorithm, the control variables are fragmented according to different model partitions as distinct local step counts exist for different parts.

---

**Algorithm 4** FedALS + SCAFFOLD

---

**Input**: Initial model $\{\boldsymbol{\theta}_{k,1,0} = [\boldsymbol{\phi}_{k,1,0}, \boldsymbol{h}_{k,1,0}]\}_{k=1}^{K}$, Initial control variable $\{\boldsymbol{c}_{k,1} = [\boldsymbol{c}_{k,1}^{\boldsymbol{\phi}}, \boldsymbol{c}_{k,1}^{\boldsymbol{h}}]\}_{k=1}^{K}$, learning rate $\eta$, number of local steps for the head model $\tau$, adaptation coefficient $\alpha$.

**Output**: $\hat{\boldsymbol{\theta}}_R$

1: **for** Round $r$ in $1, ..., R$ **do**
2:  **for** Node $k$ in $1, ..., K$ **in parallel do**
3:   **for** Local step $t$ in $0, ..., \tau - 1$ **do**
4:    Sample the batch $\mathcal{B}_{k,r,t}$ from $\mathcal{D}_k$.
5:    $\boldsymbol{\theta}_{k,r,t+1} = \boldsymbol{\theta}_{k,r,t} - \eta\big(\frac{1}{|\mathcal{B}_{k,r,t}|} \sum_{i \in \mathcal{B}_{k,r,t}} \nabla l(M_{\boldsymbol{\theta}_{k,r,t}}, \boldsymbol{z}_{k,i}) - \boldsymbol{c}_{k,r} + \frac{1}{K} \sum_{k=1}^{K} \boldsymbol{c}_{k,r}\big)$
6:    **if** $\mod (r\tau + t, \tau) = 0$ **then**
7:     $\boldsymbol{c}_{k,r}^{\boldsymbol{h}} \leftarrow \boldsymbol{c}_{k,r}^{\boldsymbol{h}} - \frac{1}{K} \sum_{k=1}^{K} \boldsymbol{c}_{k,r}^{\boldsymbol{h}} + \frac{1}{\eta\tau}(\boldsymbol{h}_{k,r,0} - \boldsymbol{h}_{k,r,t})$
8:     $\boldsymbol{h}_{k,r,t} \leftarrow \frac{1}{K} \sum_{k=1}^{K} \boldsymbol{h}_{k,r,t}$
9:    **else if** $\mod (r\tau + t, \alpha\tau) = 0$ **then**
10:     $\boldsymbol{c}_{k,r}^{\boldsymbol{\phi}} \leftarrow \boldsymbol{c}_{k,r}^{\boldsymbol{\phi}} - \frac{1}{K} \sum_{k=1}^{K} \boldsymbol{c}_{k,r}^{\boldsymbol{\phi}} + \frac{1}{\eta\alpha\tau}(\boldsymbol{\phi}_{k,\lfloor \frac{r\tau+t-\alpha\tau}{\tau} \rfloor, \mod (r\tau+t-\alpha\tau, \tau)}^{l} - \boldsymbol{\phi}_{k,r,t}^{l})$
11:     $\boldsymbol{\phi}_{k,r,t} \leftarrow \frac{1}{K} \sum_{k=1}^{K} \boldsymbol{\phi}_{k,r,t}$
12:   $\boldsymbol{c}_{k,r+1} = \boldsymbol{c}_{k,r}$
13:   $\boldsymbol{\theta}_{k,r+1,0} = \boldsymbol{\theta}_{k,r,\tau}$
14: **return** $\hat{\boldsymbol{\theta}}_R = \frac{1}{K} \sum_{k=1}^{K} \boldsymbol{\theta}_{k,R,\tau}$

