# OpenReview forum: "Communication Efficient Federated Learning with Differentiated Aggregation"
_ICML.cc/2024/Workshop/WANT — WANT@ICML 2024 Poster_

### Official Review · Reviewer_nSs2 · 2024-06-08
**Communication Efficient Federated Learning with Differentiated Aggregation**

**Confidence:** 3

**Summary:**

Below are the main contributions of the paper:

a) The paper provides a more precise generalization bound for one-round federated learning. This bound is based on the local clients' generalizations and the heterogeneity of the data distribution (non-iid scenario).

b) It extends the analysis to R-round federated learning, characterizing the generalization bound and its relationship to the number of local updates (local stochastic gradient descents (SGDs)).

c) The analysis reveals that less frequent aggregations, resulting in more local updates for the representation extractor (typically corresponding to the initial layers), lead to the creation of more generalizable models, especially in non-iid scenarios.

d) Based on the generalization bound and representation learning analysis, the paper introduces the Federated Learning with Adaptive Local Steps (FedALS) algorithm. This algorithm employs varying aggregation frequencies for different parts of the model, thereby reducing the communication cost.

e) The paper concludes with experimental results demonstrating the effectiveness of the FedALS algorithm.

**Strengths:**

Below is a summarization of the main strong points of the paper:

   - Introduces the FedALS algorithm to adapt aggregation frequencies, reducing communication overhead in federated learning.
   - Provides tighter generalization error bounds for both one-round and multi-round federated learning, considering data heterogeneity.
   - Demonstrates that less frequent aggregations for initial layers (representation extractor) lead to more generalizable models, reducing communication costs.
   - Develops FedALS, which varies aggregation frequencies for different model parts, enhancing efficiency and performance.
   - Supports theoretical contributions with experimental results showing the effectiveness of FedALS.
   - Addresses the challenge of non-iid data distributions, relevant for real-world federated learning applications.

**Weaknesses:**

- The paper's main theoretical tool which is FedALS has to be applied to more challenging real datasets that exhibit strongly non-iid behaviour such as the Clothing 1M datasets and here is a link to the dataset (https://paperswithcode.com/dataset/clothing1m).

- The related work section ignores a vast amount of related recent papers that has powerful algorithm that are very competitive to the proposed algorithm and here are few examples. The authors have to compare their results to these algorithms for fair treatment of the proposed work as it might be the case that one of these algorithms might have better performance in all aspects

 1) Mishchenko, Konstantin, et al. "Proxskip: Yes! local gradient steps provably lead to communication acceleration! finally!." International Conference on Machine Learning. PMLR, 2022.
2) Yi, Kai, et al. "Cohort Squeeze: Beyond a Single Communication Round per Cohort in Cross-Device Federated Learning." arXiv preprint arXiv:2406.01115 (2024).
3) Tyurin, Alexander, and Peter Richtárik. "A computation and communication efficient method for distributed nonconvex problems in the partial participation setting." Advances in Neural Information Processing Systems 36 (2024).

**Limitations:**

- One limitation of the proposed work is the lack of further privacy-preserving aspects in federated learning, specifically encrypting the local gradients that get uploaded to the central server while maintaining good overall performance of the learning scheme. This could be a potential future work for the authors to make the paper more comprehensive.

- Another limitation is the lack of explanation regarding the cost of optimizing the number of communication rounds in terms of the overall model performance, such as those in FedAvg. This is important because every optimization of communication rounds is likely to have some technical impact on the overall model performance.

**Suggestions:**

- I strongly recommend that the authors try freezing the initial layers of every local client after these layers learn the main representation of their training datasets. Only the gradients of the other layers should be learned and the first layers should be kept constant to speed up convergence. This approach may not compromise the overall accuracy of the final model.

---

### Decision · Program_Chairs · 2024-06-18

**Decision:**

Accept (Poster)

**Comment:**

We thank the authors for their time and contribution to WANT and we are pleased to share that after the reviewing process the paper has been accepted. Congratulations! We encourage the authors to consider reviewers' feedback for the improvement of the camera-ready version. We hope to see you in person at the workshop and brainstorm on efficient training research together!